# Numerical Analysis of the Installation Process of Screw Piles Based on the FEM-SPH Coupling Method

**Qingxu Zhao, Yuxing Wang \*, Yanqin Tang, Guofeng Ren, Zhiguo Qiu, Wenhui Luo and Zilong Ye** 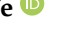

College of Engineering, South China Agricultural University, Guangzhou 510642, China
\* Correspondence: scauwyx@scau.edu.cn

**Abstract:** The installation of screw piles can cause damage to the soil, which is a dynamic and large deformation problem. In this paper, a FEM-SPH numerical model for the analysis of this large deformation problem was developed in LS-DYNA to simulate the installation process of screw piles. In addition, field installation tests of screw piles were carried out. By comparing the FEM-SPH simulation results with the experimental results and traditional FEM simulation results, it was found that the FEM-SPH coupling method has higher efficiency and accuracy in dealing with the large deformation problems caused by the installation of screw piles. Then, numerical simulations of screw piles with various parameters were conducted to analyze the differences in the installation process. The results show that the spiral pitch and pile diameter have a significant effect on the installation torque, soil stress and soil pressure during the installation process. During the installation process, the interaction between the screw pile and the soil is transferable. The installation of the screw pile will lead to the movement of soil particles in the radial and axial directions, resulting in heave damage to the shallow soil and cylindrical shear damage to the middle and deep soil. The influence range on the soil by the heave failure mechanism (HFM) and the cylindrical shear failure mechanism (CSFM) caused by the installation of screw piles is affected by pile parameters. The change in pile diameter will act on both HFM and CSFM, whereas the variation in spiral pitch will only have an influence on CSFM.

**Keywords:** screw pile; large deformation problem; installation process; pile–soil interaction; numerical simulation; FEM-SPH

## 1. Introduction

Due to the advantages in the aspects of ease and fast installation with minimal vibration and high bearing capacity [1], screw piles have been widely used in various situations [2]. In China, as a new foundation type, screw piles have been applied in the field of the foundation of solar power plants or heightening dams, slopes, landslides and so on. These piles, also known as micro screw piles, are usually hollow metal piles with continuous spiral blades, the pile lengths of which are generally between 1 and 2 m. They are mainly composed of the smooth cylindrical segment on the top, the cylindrical segment with blades in the middle and the conical tapered segment with blades at the bottom. These micro screw piles are usually screwed into the soil, which inevitably causes them to interact with the soil, bringing various problems to the installation, such as large settlement and excessive damage to the soil. The application of screw piles depends primarily on their load-bearing capacity, which in turn is influenced by the pile–soil interaction during the installation process. However, most of the research on this kind of pile focuses on the load-bearing process. Therefore, it is of great practical implication to study the installation process of screw piles.

At present, a lot of research has already studied pile–soil interaction during the installation process of screw piles, including theoretical analysis, field tests and model tests. Many scholars, by analyzing the forces on the screw pile during the installation process [3–7], or based on the installation tests of screw piles and the Cone Penetration Test





(CPT) [8–10], have proposed the theoretical calculation method of the installation torque and the resistance of screw piles. The installation of screw piles can cause damage to the soil, and some researchers have investigated this behavior of the soil. Jaafer A R et al. [11] conducted model tests using centrifugal devices and found that the soil disturbance generated during the installation process was influenced by soil density. Markou A.A and Kaynia A.M [12] introduced a generalized nonlinear mechanical formulation to solve nonlinear behavior between piles and soils and analyzed the effect of soil damping on pile–soil interaction. When Galindo, P.G et al. [13] studied the influence of rotational speed on the plugging and installation behavior of the screw pile, he found that increasing the rotational speed can significantly reduce the penetration resistance of open and closed piles during the installation process. However, pile–soil interaction is a microscopic mechanical behavior, which means the analysis of this process cannot be well completed by field experiments and theoretical analysis alone. Numerical simulation seems to provide the feasibility for this research.

Chen et al. [1] studied the effect of pile structure parameters on pile–soil interaction and load transfer under compressive loads by using a discrete element model. The finite element numerical analysis of single screw piles and pile groups in cohesion and cohesionless soils was conducted by Nowkandeh, M.J and Choobbasti, A.J to analyze the effect of soil parameters and pile structure parameters on pile–soil action and bearing capacity [14]. Alwalan M.F and El Naggar M.H [15] used the finite element method to simulate the installation of large screw piles in sandy and cohesion soils, and the factors affecting the pile–soil dynamic response were analyzed. Akopyan V and Akopyan A [16] analyzed the interaction between the screw pile and the surrounding soil by the finite element approach and derived an equation to describe the pile–soil interaction. Shi et al. [17] analyzed the effect of the drilling speed ratio (the ratio of rotational speed to descent speed) on the penetration characteristics of screw piles by the DEM model, revealing the micro-mechanical relationships between soil particles and macroscopic penetration properties of screw piles. In addition, many researchers have also investigated the pile–soil interaction and load transfer mechanisms of screw piles under different conditions at a microscopic level based on the numerical simulation method [2,18–27]. From these studies, it can be seen that the numerical simulations enable a range of factors to be considered, broadening the avenues of research. It is worth noting that most of these numerical studies on pile–soil interaction focus on compression, uplift and lateral loading processes, and little attention is paid to the installation process of screw piles. Therefore, it is necessary to investigate the pile–soil interaction during the installation process of a screw pile.

The installation of screw piles is a dynamic and large deformation problem, which means that it is difficult to solve the deformation of the meshes by traditional finite element methods. Considering the limitations of traditional finite element methods in dealing with large deformations problems, many researchers [28–32] have gradually adopted the Smoothed Particle Hydrodynamic (SPH) method to deal with large deformation problems in geotechnical engineering. Their studies [28–32] all demonstrate the suitability of the SPH method for dealing with large deformation problems.

In this paper, a FEM-SPH coupling method is introduced to investigate the installation process of screw piles. The screw pile was simulated using the finite element method (FEM), and the soil model was constructed using the SPH method. Additionally, a numerical model was established for the analysis of the installation of a screw pile in LS-DYNA. To verify the accuracy and reliability of the established model, the simulation results were compared with field test data and the traditional finite element method to demonstrate the applicability of the FEM-SPH numerical model in dealing with large soil deformations caused by screw pile installation. Then, the analysis of the screw pile installation process is presented from the aspect of the movements of surrounding soil particles at different depths. Finally, the installation simulations of different screw piles were carried out to understand the differences in pile structure parameters on the installation process.

## 2. Basic Theory of SPH

SPH is a Lagrangian smooth particle method, the underlying principle of which is to describe the continuum by discretizing it into interacting individual particles. Each particle carries physical quantities associated with itself, such as mass and velocity. In the process of solving it, the motion state of the whole continuum can be obtained by solving the kinetic equations of the particles group and tracking the motion of each particle. The conservation equations can be expressed in terms of fluxes or internal force of the particle.

### 2.1. Solution of the SPH

The core of the SPH method is interpolation, and an important step in the computation is the domain search. As is shown in Figure 1, this search is carried out on the spherical region with a radius of 2 *h* (smoothing length) in the influence domain of each particle, and all particles in this region are listed for comparison at each time step. At the same time, the smooth length *h* is used to ensure that there are sufficient influence domains within the critical domain of a particle to avoid problems with material compression and expansion. When the SPH particles are separated from each other, the smooth length will increase. Further, when particles are moved closer to each other, the smooth length decreases. This can ensure that the same number of particles are contained in the critical domain to complete the calculation.

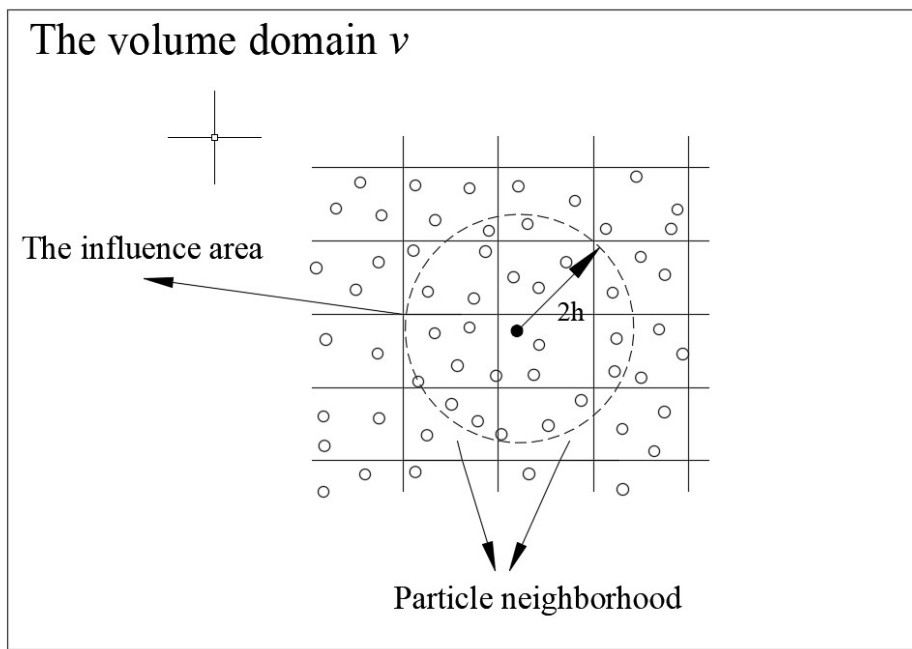

**Figure 1.** The domain search of the SPH method.

For the solution of the SPH method, it is usually divided into two steps. The first step is to obtain the field function in the continuous form as an integral representation (Kernel function). The second step is to approximate the SPH particles to achieve the particle discretization of the integral expression.

In the derivation of an SPH formulation, the coordinate integral form of the particle is usually expressed as:

$$F(x) = \int_v F(x')\partial(x - x')dx' \tag{1}$$

where $v$ is volume domain, $x$ is coordinate vectors of particle, $x'$ is coordinate vectors of any particle in the volume domain and $\partial(x - x')$ is the Dirac delta function [29], and it can be given as:

$$\partial(x - x') = \begin{cases} 1, & x = x' \\ 0, & x \neq x' \end{cases} \tag{2}$$

In fact, the Dirac delta function is difficult to realize because it has only one non-zero value. It is usually replaced by the smoothing function $W(x - x', h)$. Therefore, the coordinate integral form of the particle can also be expressed as:

$$F(x) = \int_v F(x')W(x - x', h)dx' \tag{3}$$

where $h$ is the smoothing length.

The smoothing function is usually expressed by $B$ spline smooth functions. It can be expressed as:

$$W(x - x', h) = W(q, h) = \alpha_q \times \begin{cases} \frac{2}{3} - q^2 + \frac{q^3}{2} & 0 \le q < 1 \\ \frac{(2-q)^2}{6} & 1 \le q < 2 \\ 0 & q \ge 2 \end{cases} \tag{4}$$

where $q = |x - x'|$ is the distance between particles and $\alpha_q$ is $\frac{15}{7\pi h^2}$ and $\frac{3}{2\pi h^2}$ in two-dimensional space and three-dimensional space.

The smoothing function $W$ is related to the smooth length $h$ of the particle influence domain and the corresponding position between the particles.

According to the definition of the SPH method, the continuum, when is discretized by the SPH method, is a number of particles with a certain mass and volume. Thus, for the second step of the solution of the SPH, the physical quantities of the particle can be summed up by the physical quantities of all particles in the particle's influence area. This is also an important reason for the discrete SPH particles.

The SPH method is based on a quadrature formula for moving particles, so the particle approximation of $f(x)$ can be expressed as:

$$f(x) \approx \sum_{j=1}^{M} \frac{m_j}{\rho_j} f(x_j) W(x - x_j, h) \tag{5}$$

Therefore, the function of the particle $i$ is approximated as:

$$f(x_i) \approx \sum_{j=1}^{M} \frac{m_j}{\rho_j} f(x_j) W(x_i - x_j, h) \tag{6}$$

where $\rho_j$ is the density of particle $j$ and $m_j$ is the mass of particle $j$. $M$ is total number of particles in the influence area at $x$.

According to Equation (6), the particle approximation of the derivative of $f(x)$ can be obtained:

$$\frac{\partial f(x)}{\partial x} = \sum_{j=1}^{M} \frac{m_j}{\rho_j} f(x_j) \frac{\partial W(x - x_j, h)}{\partial x} \tag{7}$$

Therefore, it can be converted to use the SPH method to find the physical quantity at any position $m$; the equation can be given as:

$$A(x_m) = \frac{\partial f(x_m)}{\partial x} = \sum_{j=1}^{M} \frac{m_j}{\rho_j} f(x_j) \frac{\partial W(x_m - x_j, h)}{\partial x} \tag{8}$$

where $A(x_m)$ is a scalar quantity and can be any property of the continuum, including density, cohesion and so on.

## 2.2. FEM-SPH Coupling Algorithm

The contact problem of FEM-SPH is the key to the successful calculation of the numerical model. Presently, the methods to deal with the contact between finite element and SPH particles are mainly the kinematic constraint method and the penalty method. In most

coupled contacts, the penalty function is one of the most commonly used methods [29]. During the FEM-SPH coupling process in this paper, it is important to ensure that no penetration between the FE screw pile and SPH particles of the soil occurs. Therefore, to prevent penetration, the contact between finite elements and SPH particles is defined using the penalty function; the contact model is shown in Figure 2. The contact boundary needs to satisfy the following equation:

$$\left(\frac{\partial}{\partial t}\boldsymbol{u} - \boldsymbol{s}\right)\cdot\boldsymbol{c} = 0 \tag{9}$$

where $\boldsymbol{u}$, $\boldsymbol{s}$ and $\boldsymbol{c}$ are the displacement of the element nodes at the boundary, the velocity of the SPH particles at the boundary and the normal vectors of the elements at the boundary, respectively.

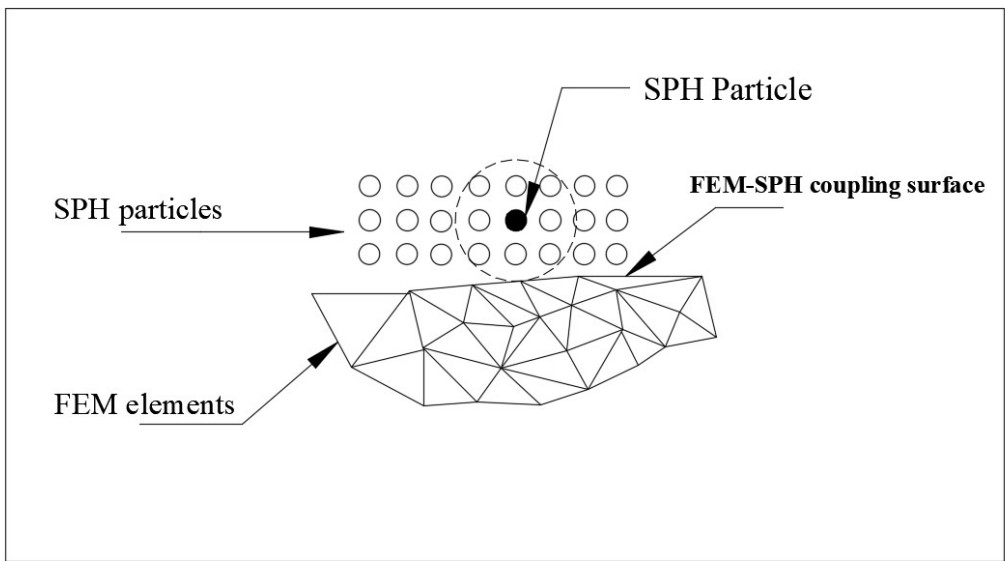

**Figure 2.** The contact model between FEM and SPH.

According to the studies of Christopher Goodin [28], the mass conservation equation and the momentum conservation equation of particles can be described below, respectively:

$$\frac{d\rho_i}{dt} = \rho_i \sum_j^M \frac{m_j}{\rho_j}(\boldsymbol{U}_j - \boldsymbol{U}_i)\frac{\partial W(\boldsymbol{x}_i - \boldsymbol{x}_j,\ h)}{\partial x} \tag{10}$$

$$\frac{d\boldsymbol{U}_i}{dt} = \sum_j^M m_i\left(\frac{z_j}{\rho_j} + \frac{z_j}{\rho_j} + C_{ij} + \sigma_{ij}\right)\frac{\partial W(\boldsymbol{x}_i - \boldsymbol{x}_j,\ h)}{\partial x} \tag{11}$$

where $\boldsymbol{U}$ is the soil velocity, $z$ is stress tensor, $C_{ij}$ is the artificial viscosity and $\sigma_{ij}$ is the artificial stress terms.

According to the previous continuity equation after the SPH discretization, the momentum and mass conservation equations, the volume force $T(\boldsymbol{x}_i)$ and inter-particle contact force $F(\boldsymbol{x}_i)$ of the particles at $\boldsymbol{x}$ can be obtained as follows:

$$T(\boldsymbol{x}_i) = \sum_{j=1}^{N_a+N_b} \frac{m_j}{\rho_j} Kn \frac{W(\boldsymbol{x}_i - \boldsymbol{x}_j)^{n-1}}{W(h_{avg})}\frac{\partial W(\boldsymbol{x}_i - \boldsymbol{x}_j,\ h)}{\partial x} \tag{12}$$

$$F(\boldsymbol{x}_i) = \sum_j^{N_a+N_b} \frac{m_j}{\rho_j}\frac{m_i}{\rho_i} Kn \frac{W(\boldsymbol{x}_i - \boldsymbol{x}_j)^{n-1}}{W(h_{avg})^n}\frac{\partial W(\boldsymbol{x}_i - \boldsymbol{x}_j,\ h)}{\partial x} \tag{13}$$

where $K, n$ are scalars defined by the contact interface penalty function, $h_{avg}$ is the average of the distance between particle $i$ and particle $j$ and $N_a$ and $N_b$ are the number of SPH particles and FEM nodes in the influence domain.

The contact between the screw pile and the soil is a point–surface contact, where the SPH particles of the soil model are slave contact points and the FE nodes of the pile model are master contact surfaces. According to the force interactions, the nodal forces of the SPH particles are distributed to the master nodes of the screw pile. Therefore, the total contact force $F(t)_{interface1}$ generated between the screw pile and the soil can be synthesized from the contact force generated by the SPH particles in contact with the screw pile by the following equation:

$$F(t)_{interface1} = \sum_{i=1}^{N_s} F_r(t) \tag{14}$$

where $F_r(t)$ is the inter-particle contact force of SPH particles, and $N_S$ is the total number of SPH particles in contact with the screw pile during the installation process, which varies with the time.

For the FE screw pile, a tetrahedral element is adopted to divide the grid, and the total contact force of SPH particles will act on the FE nodes, which can be expressed as:

$$F_e(t) = \frac{2W_k F_r(t)}{1 + w_{\frac{2}{3}} + w_2^2 + w_1^2} \tag{15}$$

where $W_k$ is the center coordinate of the contact point between SPH particles and elements relative to the triangular element surface.

Therefore, the contact force subjected by the FE elements can be calculated through accumulation:

$$F(t)_{interface2} = \sum_{j=1}^{N_m} F_e(t) \tag{16}$$

where $N_m$ is the total number of FE nodes in contact with SPH particles of the soil.

Therefore, for the numerical FEM-SPH coupling model, the torque during the installation process can be expressed as:

$$T(t) = \sum_{e=1}^{N_m} F_e(t) \times r_e(t) \tag{17}$$

where $r_e$ is the vector distance between the finite element node and the central axis of the screw pile.

### 3. Numerical Simulations

The installation of screw piles is a dynamic and large deformation problem, which means that it is difficult to analyze the installation process of screw piles by theoretical analysis or field tests alone. Compared to the theoretical analysis or field tests, numerical simulation seems to be able to reproduce the installation of the screw pile in a realistic way. In this section, as shown in Figure 3, a coupled FEM-SPH model was developed for simulating the installation process of screw piles in LS-DYNA. In order to verify the reasonability of the SPH soil model, the established soil model was tested using CPT numerical tests. In addition, in order to verify the accuracy of the FEM-SPH coupled model, the calculated results of the FEM-SPH method were compared with the calculated results of the conventional finite element method (Arbitrary Lagrangian–Eulerian Formulation) and field test data.

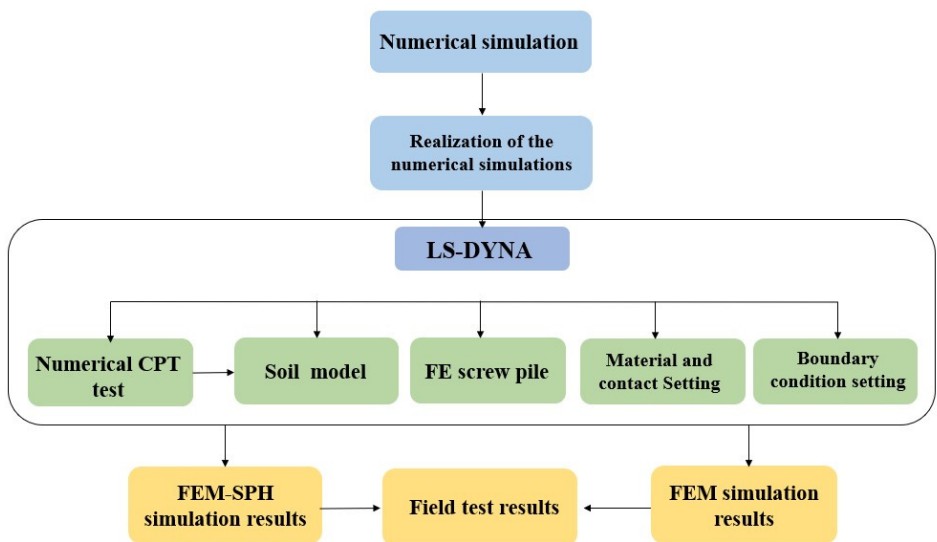

**Figure 3.** The idea of numerical simulations.

### 3.1. Numerical Model

In order to verify the numerical model, the field tests of screw piles were carried out. The field tests were carried out at the South China Agricultural University in Guangzhou. The soil of Guangzhou is latosol. The description and relevant characteristics of the soil are shown in Table 1. As is shown in Figure 4, field tests of four types of screw piles were carried out to measure the installation torque; the structural parameters of the test screw piles are shown in Table 2. The test pile is referred to as "TP"; for example, "TP2" indicates field test 2, namely, the screw pile with the pile diameter 90 mm and spiral pitch 100 mm. All screw piles were instrumented with strain gauges to measure the torque. During the field test, ensure that all screw piles are installed to a depth of 700 mm to reduce errors during measurement. In this paper, the field test data of the installation torque were compared with the simulation results.

As shown in Figure 5, the key steps of the numerical simulation are listed. As for the establishment of the numerical model, considering the small deformation of the screw pile, we adopted the finite element method (FEM) to establish the model of screw piles. The models were developed to match the full size of the test piles, and the smaller size (0.007 m) was used for meshing to improve the computational accuracy of the numerical model. According to the complexity of the structure in the model, the grid type of the screw pile was determined as tetrahedron; the FE screw pile model is shown in Figure 6a. The screw piles are made of Q235 steel (the keyword *MAT_RIGID), the material parameters and as shown in Table 3. In addition, a rotational speed was applied to the screw pile (the keyword *BOUNDARY_PRESCRIBED_MOTION_RIGID), ensuring that it could only move along the vertical direction (Z-axial) so as to simulate the actual installation of the screw pile. To avoid the confusion between the field test and numerical simulation, numerical simulations are referred to as "NT"; for example, "NT2" indicates numerical simulation2, namely, the test screw pile TP2.

**Table 1.** Soil parameters.

| $\rho$ (kg/m$^3$) | $c$ (kPa) | Poisson's Ratio $\mu$ | $E$ (kPa) | $G$ (MPa) | $K$ (MPa) |
|---|---|---|---|---|---|
| 1850 | 20 | 0.3 | 20 | 7.7 | 16.7 |

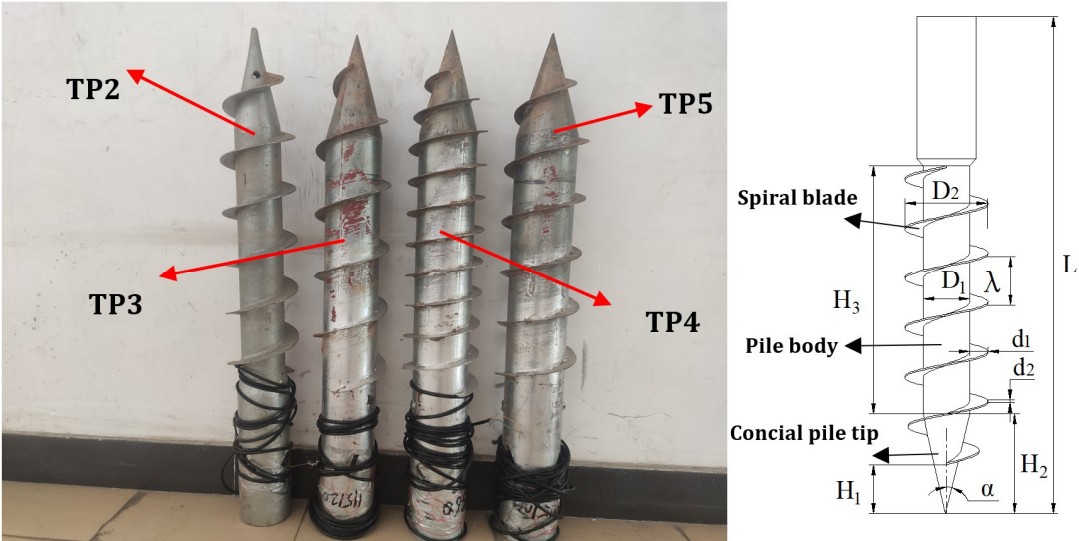

**Figure 4.** The test screw piles.

**Table 2.** Structural parameters of the test screw piles.

| Number | $L$ (mm) | $D_1$ (mm) | $D_2$ (mm) | $\lambda$ (mm) | $H_2$ (mm) | $H_3$ (mm) | $H_1$ (mm) | $d_2$ (mm) | $d_1$ (mm) |
|--------|---------|-----------|-----------|--------------|-----------|-----------|-----------|-----------|-----------|
| TP2 | 1000 | 90 | 150 | 100 | 200 | 500 | 50 | 3 | 30 |
| TP3 | 1000 | 115 | 175 | 100 | 200 | 500 | 50 | 3 | 30 |
| TP4 | 1000 | 115 | 165 | 60 | 200 | 500 | 50 | 3 | 25 |
| TP5 | 1000 | 115 | 165 | 120 | 200 | 500 | 50 | 3 | 25 |

Compared to the screw pile, the soil had a large deformation and the SPH method was adopted to develop the soil model in LS-DYNA. In addition, the conventional finite element method (Arbitrary Lagrangian–Eulerian Formulation) was also adopted to establish the soil model to compare the applicability with the FEM-SPH method. According to the research of Wang [29], in the numerical simulation of the SPH method, with the reduction in the SPH particles' spacing, the particle splash phenomenon was improved. Therefore, as is shown in Figure 6b, the smaller particle spacing (0.01 mm) was used to establish the soil model. In the numerical simulation, the soil parameters of the test site are applied to the SPH soil model (the keyword *MAT_SOIL_AND_FORM). As shown in Figure 5, for the contact between the screw pile and the soil, the FEM-SPH method adopts point–surface contact (the keyword CONTACT_AUTOMATIC_NODE_TO_SURFACE), and the FEM method (ALE) uses the CONSTRAINED_LAGANGE_IN_SOILD keyword to define the contact. In addition, the soil is simulated as isotropic, and the influence of pore pressure on the soil model is not considered in numerical simulation.

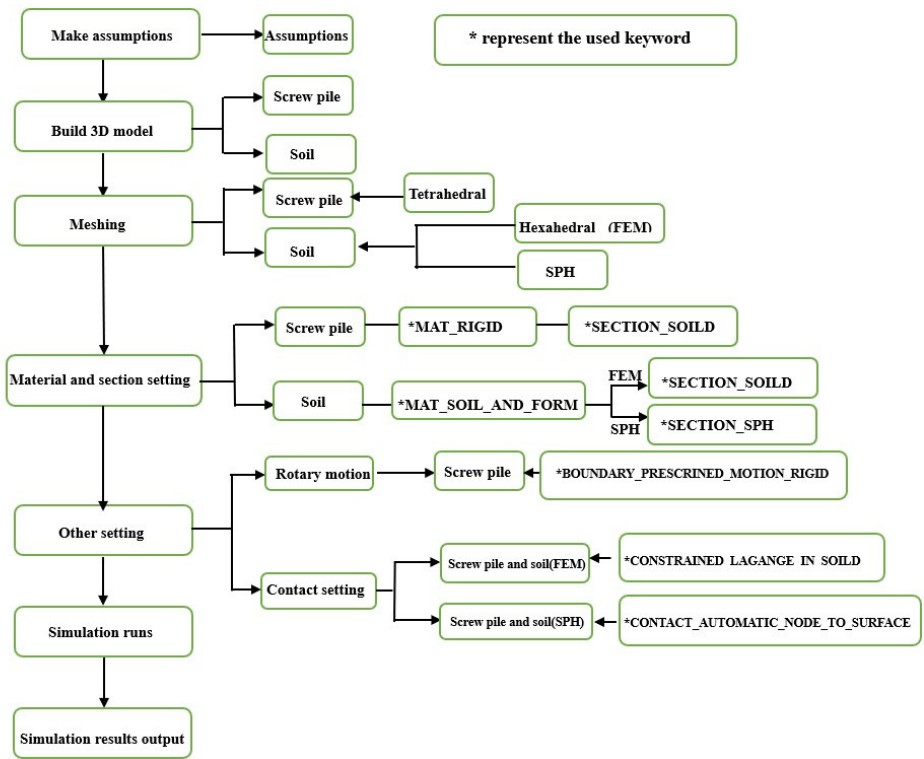

**Figure 5.** Flow chart of simulation.

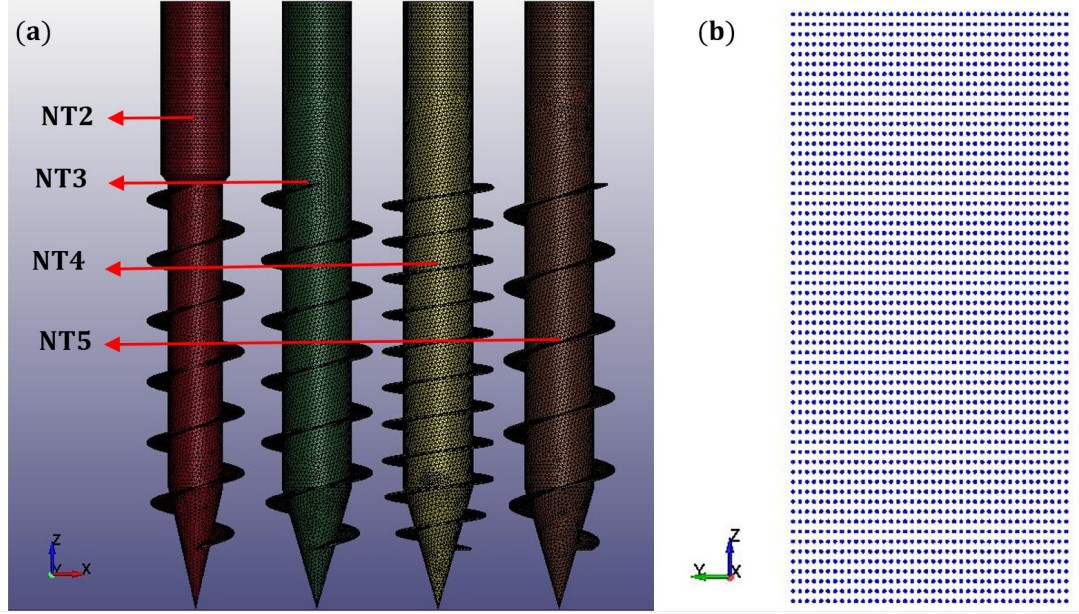

**Figure 6.** Numerical models: (**a**) FE screw pile model, (**b**) SPH soil model.

**Table 3.** Material parameters of screw piles.

| Material | $\rho$ (kg/m$^3$) | $E$ (MPa) | Poisson's Ratio $\mu$ |
| --- | --- | --- | --- |
| Q235 | 7800 | $2.1 \times 10^5$ | 0.3 |

### 3.2. Influence of the Size of the Soil Model

In numerical simulations or model tests, the minimum spacing between the screw pile and container walls was greater than 10 times the largest helix diameter to ensure no boundary effects [33]. In fact, during the numerical simulation, we established two sizes of the soil model; the spacing between the screw pile and boundary was two and four times the helix diameter. In the calculations, only the size of the soil models is different, and all other settings and parameters are the same. After a preliminary calculation of the installation process of the screw pile, the simulation results were shown in Figure 7. It can be seen at the end of the screw pile installation, the soil damage range of the two models showed consistency and neither extended to the boundary. Given the scale and time of the numerical model calculations, the actual boundary ranges of the soil models are not as they are suggested [33].

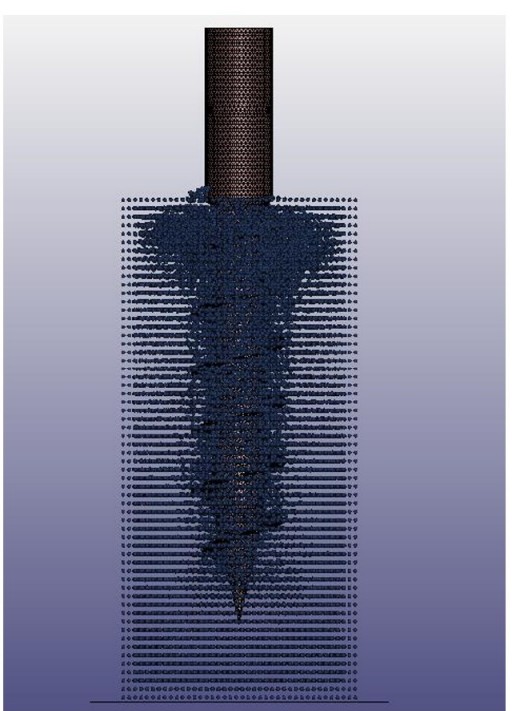 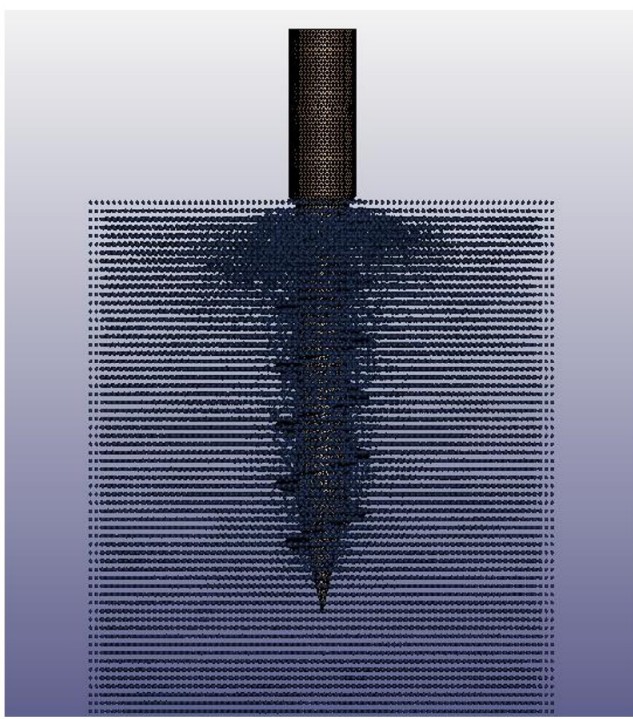

**Figure 7.** The range of soil disturbance for different soil models after the installation of the screw pile.

### 3.3. Verification of the Soil Model

For the soil model built using the SPH method, it is necessary to judge whether the model is reasonable. Cone Penetration Test (CPT) is an in situ test method that has been widely used to measure the corresponding soil properties and obtain continuous soil profile information [34]. As shown in Figure 8a, a numerical CPT test was carried out on the soil model built using the SPH method, and the penetration resistance of the cone at different penetration speeds was measured. The test results are shown in Figure 8b; it can be seen that as the installation depth increases, the penetration resistance of the cone increases, but the rate of growth of its resistance decreases. The speed of penetration has a greater influence on the penetration resistance. The greater the speed of the cone during the penetration process, the greater the penetration resistance. The numerical simulation results are consistent with the discrete element soil model established by Li et al. [35] and the penetration results of the CPT test conducted by Li et al. [36], indicating that the soil model established using the SPH method can reflect the real soil properties.

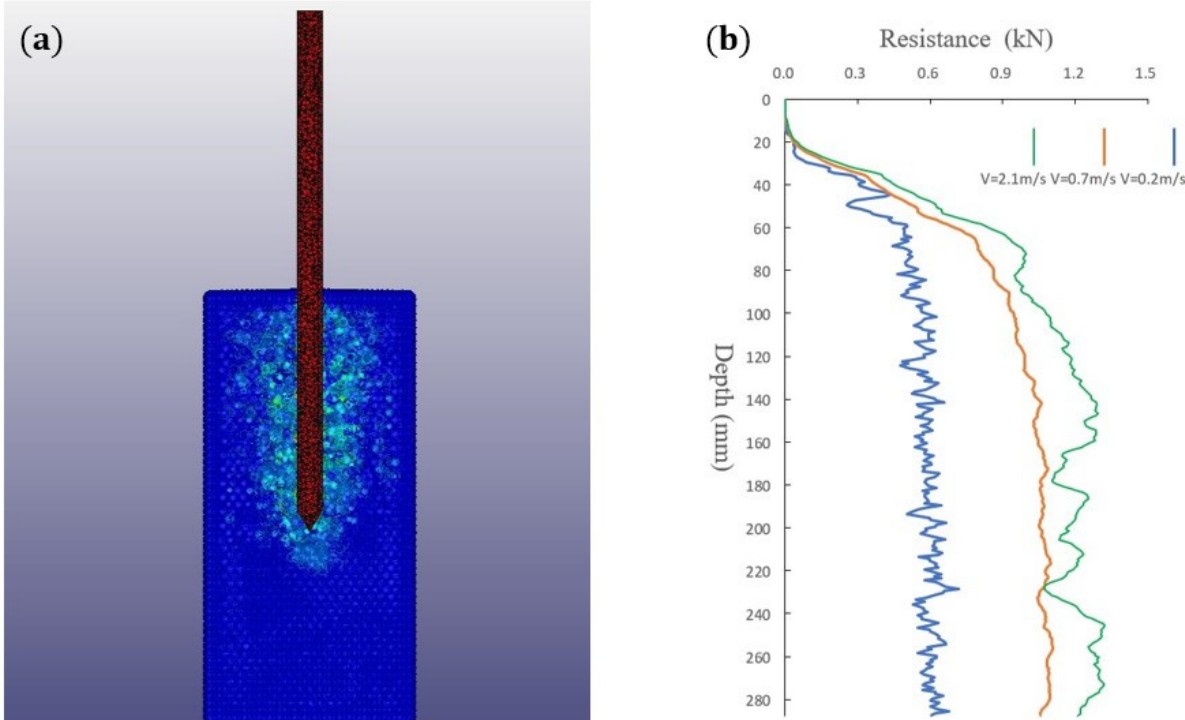

**Figure 8.** CPT test results: (**a**) distribution of soil displacement, (**b**) development of penetration resistance at different velocity.

### 3.4. Verification of the FEM-SPH Model

In this section, in order to verify the reasonability of the established numerical model, as is shown in Figure 9, the results of the FEM-SPH numerical model were compared with the experimental results. It can be found that the overall trend of the torque-installation depth curves obtained from test data and numerical simulation results is consistent. All the installation torque obtained from field tests and numerical simulation increases with the increasing depth and reaches a maximum value at the end of the installation. The trend of curves is quite consistent with the conclusion of Ghaly A and Skar [3,7]. In addition, the installation torque is related to the pile structural parameters; it is positively correlated with the pile diameter and negatively related with the spiral pitch. This phenomenon can also be seen in Figure 9a,b, where, for all numerical simulations, the trend of all curves follows a similar pattern to the field data. A comparison of the maximum torque of all piles between the experiments and numerical simulation is further given in Figure 9c. It can be seen that all the simulation results are smaller than the experimental data, but the error in both cases is within 20%. Given the improper handling of the tests and the diversity of soil conditions, the simulation results can be considered to be in acceptable agreement with the test results. This demonstrates the validity of the numerical model developed using the coupled FEM-SPH approach.

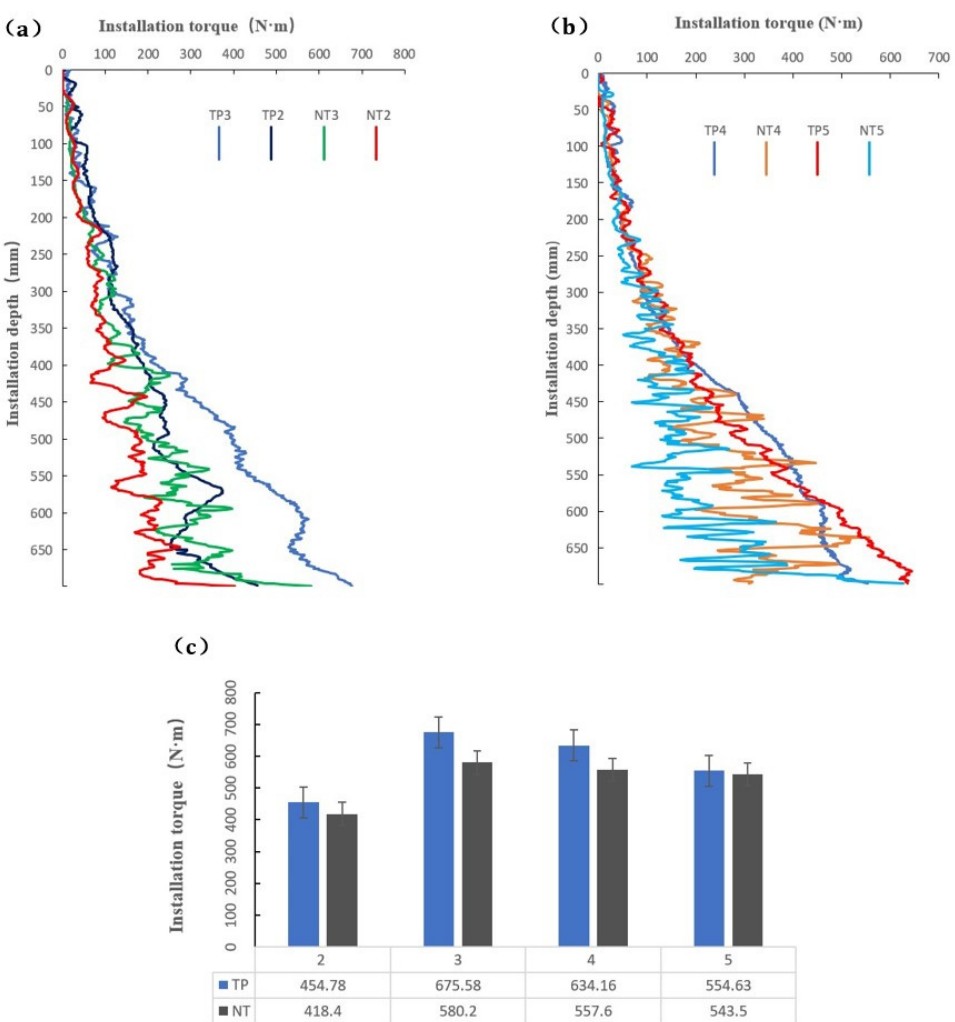

**Figure 9.** Comparison of test results with numerical simulation results: (**a**) screw piles with different diameter; (**b**) screw piles with different pitch; (**c**) the Max installation torque of four screw pile.

*3.5. Comparison between the FEM-SPH Method and FEM*

In this section, the installation of NT2 was simulated using the conventional FEM method and the FEM-SPH method, respectively, and a comparison between the field test data and two simulation results was made, including the ground soil deformation and the maximum installation torque. All two numerical models used the same settings and material parameters as the experiments. At the end of the installation, the ground soil deformation of two numerical models and the field test is presented in Figure 10. As in the field test (Figure 10a), it can be seen that the ground soil deformation was observed in both the FEM-SPH (Figure 10b) and FEM simulation (Figure 10c,d), with ground uplift occurring in both cases. In terms of the ground soil deformation, both the FEM-SPH coupling method and FE method can reflect the consistency with the field test.

The maximum installation torque obtained from the two different numerical simulations and field test is presented in Figure 11. The results show that the max torque in the FEM model and the FEM-SPH model is 418.4 N m and 497.8 N m, meanwhile, the experimental data are 454.7 N m. The errors between the two different numerical simulations and the experimental results are 7.9% and 8.65%, respectively. In terms of the max installation torque, the FEM-SPH coupling method is more consistent with field tests than the finite element method. For the FEM and FEM-SPH numerical models, the CPU calculation time to complete the simulation of the screw pile installation is shown in Figure 11. It can be found that the calculation time for the simulation of the FEM method is greater than that

of the FEM-SPH method. In terms of computational efficiency, the FEM-SPH coupling method is also greater than the finite element method.

Figure 12 shows the variation in soil stresses in the two numerical simulations at the end of the installation. It can be seen that the region of soil stress variation in both numerical models is in the vicinity of the pile body and the spiral blades. In practice, as the screw pile is screwed into the soil for installation, this will have as little effect as possible on the deep soil. However, in the finite element simulation, the soil below the screw pile showed a significant stress concentration, which is not consistent with the actual situation. From the above analysis, it can be seen that using the FEM-SPH method to simulate the installation process of screw piles is a good choice and has good applicability for dealing with large installation deformations.

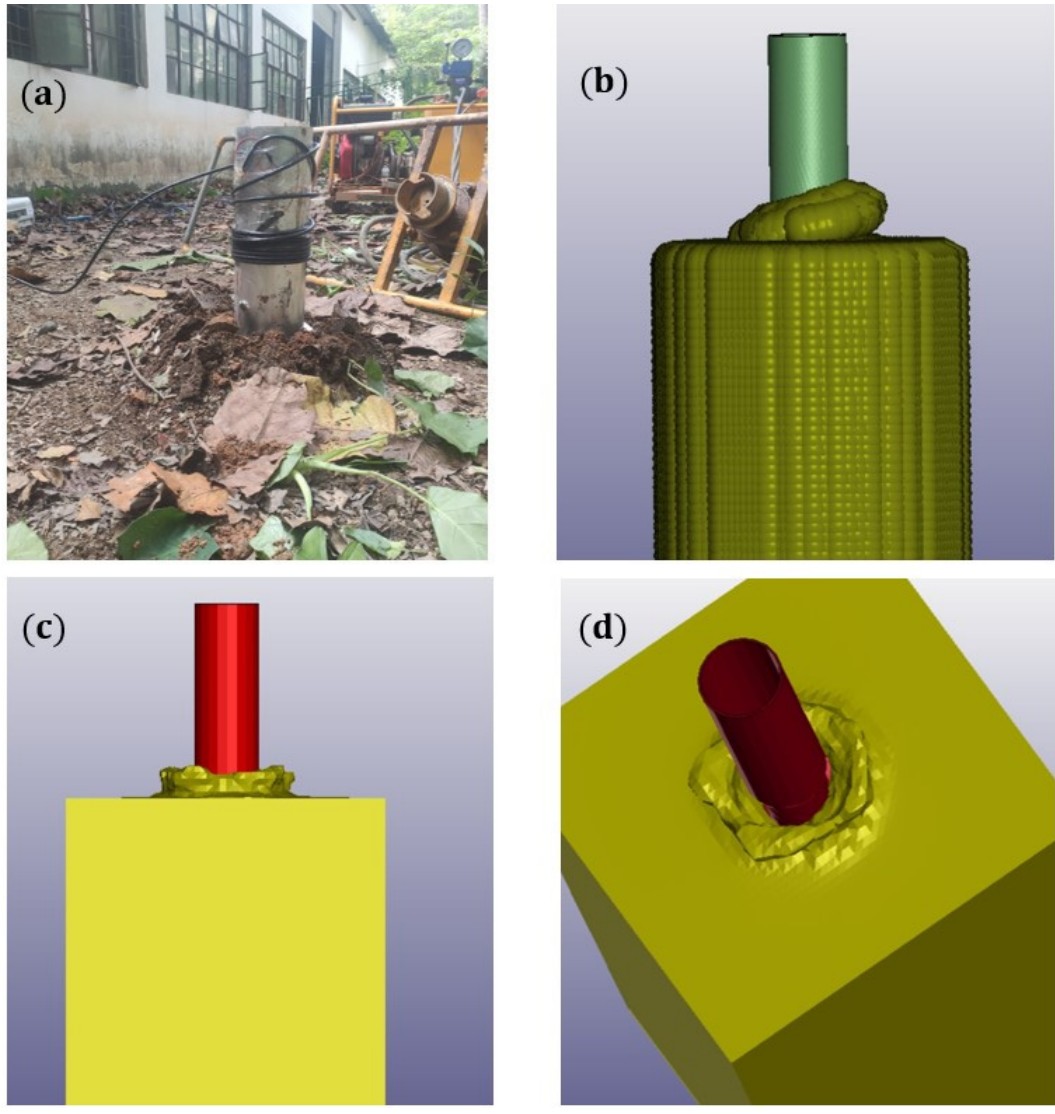

**Figure 10.** Comparison of the soil deformation between numerical simulation and field test: (**a**) field test; (**b**) FEM-SPH; (**c**,**d**) FEM.

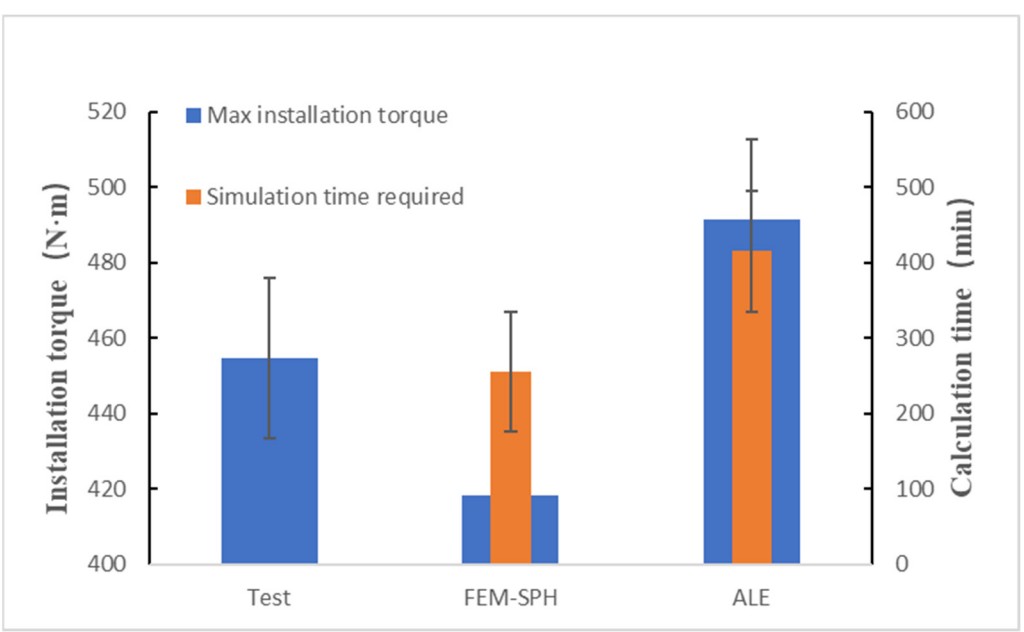

**Figure 11.** Comparison between numerical simulations results and the field test.

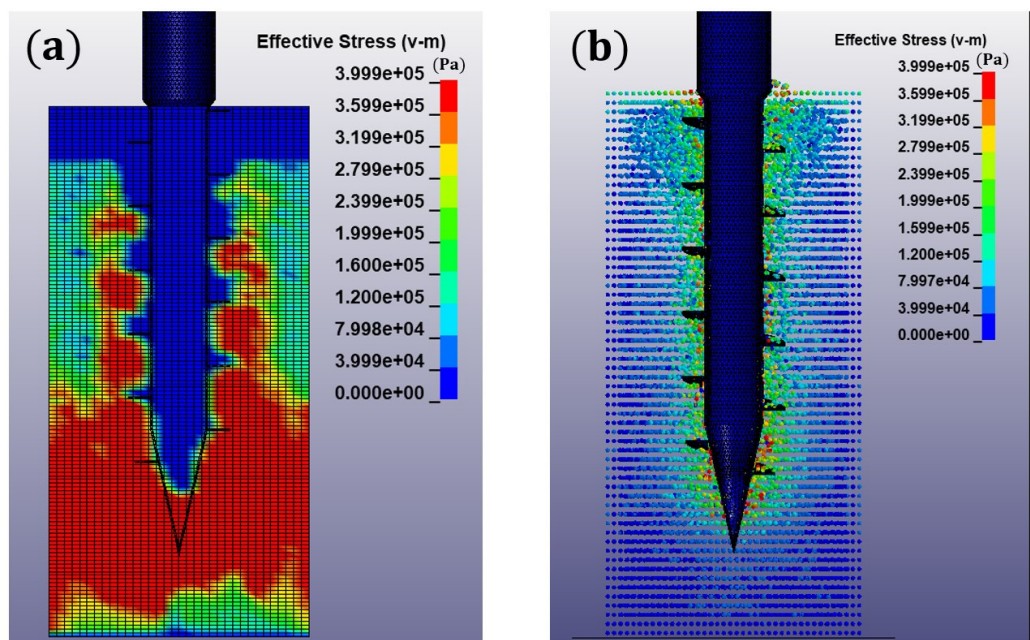

**Figure 12.** The variation in soil stresses around the screw pile: (**a**) FEM and (**b**) FEM-SPH.

## 4. Numerical Results

In this section, numerical results are given from the aspects of the movement displacement of soil particles around the screw pile at different depths to analyze the installation process of screw piles. It is worth noting that in the practical application of screw piles, the different application situations determine the variation in pile types. Therefore, it is necessary to understand the influence behavior on the surrounding soil around the screw pile during the installation process of different screw piles. All numerical simulations were based on the established FEM-SPH model mentioned in Section 3.1.

### 4.1. Analysis of the Installation Process of Screw Piles

During the installation process, the penetration of screw piles can cause damage to the soil. However, for the installation tests of screw piles, the soil deformation can be usually observed at a macroscopic level, and it is hard to measure the specific movement displacement of the soil. During the FEM-SPH numerical simulation, as is shown in Figure 13, this can be achieved by setting a series of measurement areas to obtain the movement displacement of the soil particles around the screw pile at different locations during the installation process. Due to the idealization and simplification of the numerical model, the focus is on the qualitative analysis of the installation process through the movement displacement of these soil particles. The following is the analysis of the screw pile installation process using NT3 as an example.

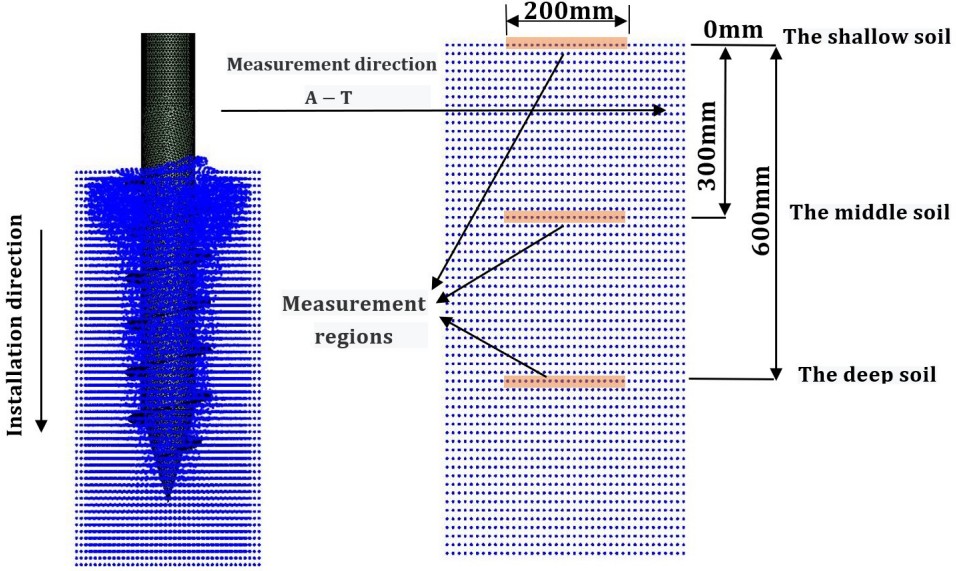

**Figure 13.** Arrangement of measurement regions.

### 4.1.1. Analysis of the Installation Process in the Shallow Soil

Figure 14 shows the movement displacement curves of the shallow soil particles (0 mm) during the installation of NT3. It can be seen that the soil particles start to move from the pile–soil contact (Time = 0 s) in the radial or axial direction, indicating that the movement of soil particles can only occur under the action of the screw pile during the installation process. For the movement displacement of soil particles, its movement is not unlimited, rather, it will gradually become stable after moving to a certain distance. As is shown in Figure 14a, it can be seen that the displacement curves of the shallow soil particles in the radial direction exist in three states. In the first stage, the displacements of soil particles increase in a straight line with infinite slope in a short period of time. Then, the displacement growth rate becomes relatively slow and eventually reaches a stable value. In the third stage, the displacement of soil particles reaches a stable state, and there will be no change.

The change in the soil particle's motion is related to this screw pile structure. In the first stage, the movement of soil particles is mainly caused by the rotational squeezing of the pile tip (H1) without spiral blades. Due to the penetration of the screw pile, the displacement of soil particles begins to increase sharply. In the second stage, as is shown in Figure 14a, the radial displacement of the soil particles can also continue to increase when the screw blade contacts the soil (Time = 0.072 s). As is shown in Figure 14a, when the conical pile tip is completely installed (Time = 0.385 s), the motion displacement of soil particles basically reaches the steady state. From the above analysis, it can be seen that the movement of soil particles is mainly caused by the installation of the conical pile tip. This is mainly because the blades of this screw pile are continuous and the spiral pitch is equal; when the pile tip is

completely installed, the blades of the pile body will be installed along the same trajectory, which makes the smallest damage to the soil, so that the motion of soil particles no longer occurs. The movement of soil particles is related to the load transfer between the screw pile and the soil. It can be seen from Figure 14a,b that all soil particles (at positions J, K, L, M and N) with larger motion displacement are located around the screw pile, indicating that the strength of load transfer is related to the distance between soil particles and the screw pile. According to the selected measurement area, it shows that the installation of NT3 has a minimal effect on the soil outside the selected location, further indicating that there is no boundary effect or a small boundary effect of the established soil model.

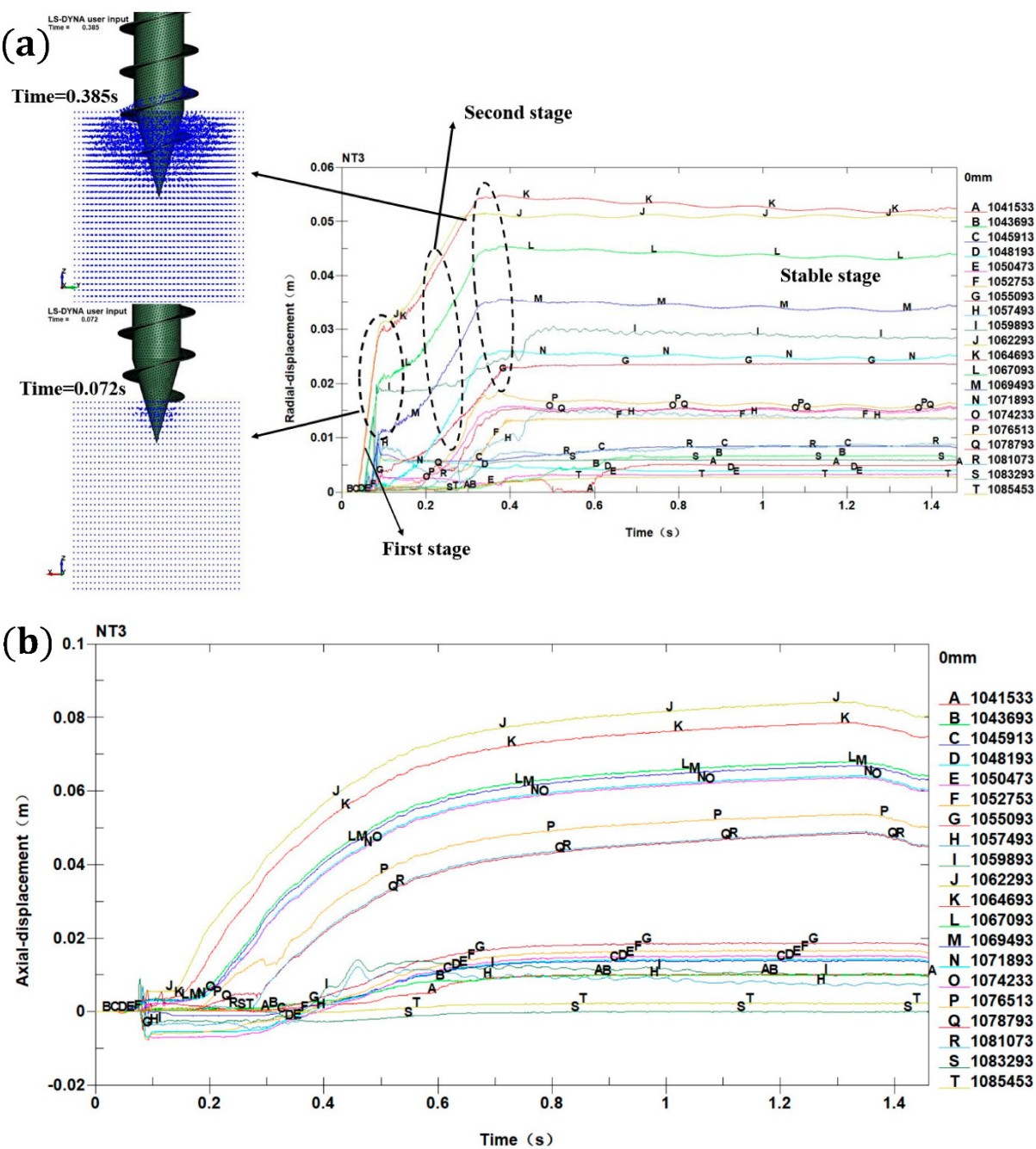

**Figure 14.** Displacement of the shallow soil particles in different directions: (**a**) radial and (**b**) axial.

The movement displacement curves of soil particles in the axial direction are shown in Figure 14b. It can be found that the movement direction of soil particles is along the

positive Z-axis, which is opposite to the direction of screw pile installation. This is mainly because the shallow soil is relatively loose and the overburden pressure is small, which makes surface heave occur after the installation of the screw pile. This reflects the heave failure mechanism (HFM) to the shallow soil during the installation of the screw piles.

### 4.1.2. Analysis of the Installation Process in the Middle Soil

Figure 15 shows the movement displacement curves of the middle soil particles (300 mm) during the installation of NT3. It can be found that the middle (300) soil particles begin to move at the same time as the shallow soil particles during the installation process, indicating that the effect of the screw pile installation on the shallow soil also influences the motion of the middle soil particles. This shows the transferability of the pile–soil interaction during the installation process. As shown in Figure 15a, it can be found that when the installation depth of the screw pile is 300 mm (Time = 0.636), the soil particles in the middle layer begin to move violently. At this moment, the conical pile tip starts to contact the soil in the middle layer (300), combined with the reasons for the movement of soil particles occurring at the surface, indicating that the violent movement of soil particles here is also caused by the installation of the conical pile tip. Unlike the shallow soil particles' displacement curves, the motion displacement curves of the middle soil particles exhibited a single linear increase in the radial direction, indicating that the soil is influenced differently by the installation of the screw piles at different depths.

As shown in Figure 15b, there is a consistency in the beginning time and duration of the violent motion of soil particles in the axial and radial directions, indicating that the motion of the soil particles in the radial and axial directions occurs simultaneously during the installation process. Compared with the motion direction of the shallow soil particles, the middle soil particles exhibit motion displacement in both directions (Figure 15b). Therefore, it can be concluded that the motion of the middle soil particles does not show a continuous overall downward or upward movement during the installation process, rather, the movement reaches a stable state after a certain displacement. With the increase in installation depth, especially after the installation of the tapered pile tip is completed at this depth, the middle soil particles will gradually reach a stable state. Compared with the shallow soil particles' displacements, the motion displacements of the middle soil particles are generally smaller in both radial and axial directions. This indicates that the disturbance displacement of the soil caused by the installation of the screw pile will gradually decrease with the increase in the installation depth. Due to the existence of soil overburden pressure and the action of the spiral blades, unlike the heave failure mechanism in the shallow soil, the installation of the screw pile exhibits a cylindrical shear failure mechanism (CSFM) to the middle soil.

### 4.1.3. Analysis of the Installation Process in the Deep Soil

As is show in Figure 16a, the motion displacement curves of the deep soil particles can be roughly divided into two parts: the continuous stable state and the irregular motion phase. In the first stage, all soil particles undergo a slight movement and then maintain a relatively stable state. According to the beginning time of soil particles motion, it can be found that the motion of the deep soil particles (600 mm) will be affected once the screw pile starts to contact the soil during the installation process. This indicates that the load transfer between the pile and the soil occurs throughout the installation of the screw pile. In the second stage, the movement curves of soil particles showed obvious irregularity, which occurs at the terminal phase of the installation of the screw pile and last until the completion of the screw pile's installation (Time = 1.46 s). As shown in Figure 16a, the installation depth of the screw pile is 600 mm, at this time the conical pile tip and soil particles began to contact each other (Time = 1.304 s), resulting in the further occurrence of disorderly movement of soil particles here, indicating that the installation of the pile tip (H1) without spiral blades is the main reason for the movement of deep soil particles.

It can be seen from Figure 16b that the beginning time of soil particles' movement in the axial is the same as the radial direction, and the movement displacement of the deep soil particles is smaller than that of the shallow and middle soil particles, indicating that the installation of screw piles has a lower effect on the disturbance of the deep soil. The direction of motion of deep soil particles in the axial direction is the same as that of the middle soil particles, reflecting the cylindrical shear failure mechanism caused by the installation of the screw pile on the deep soil.

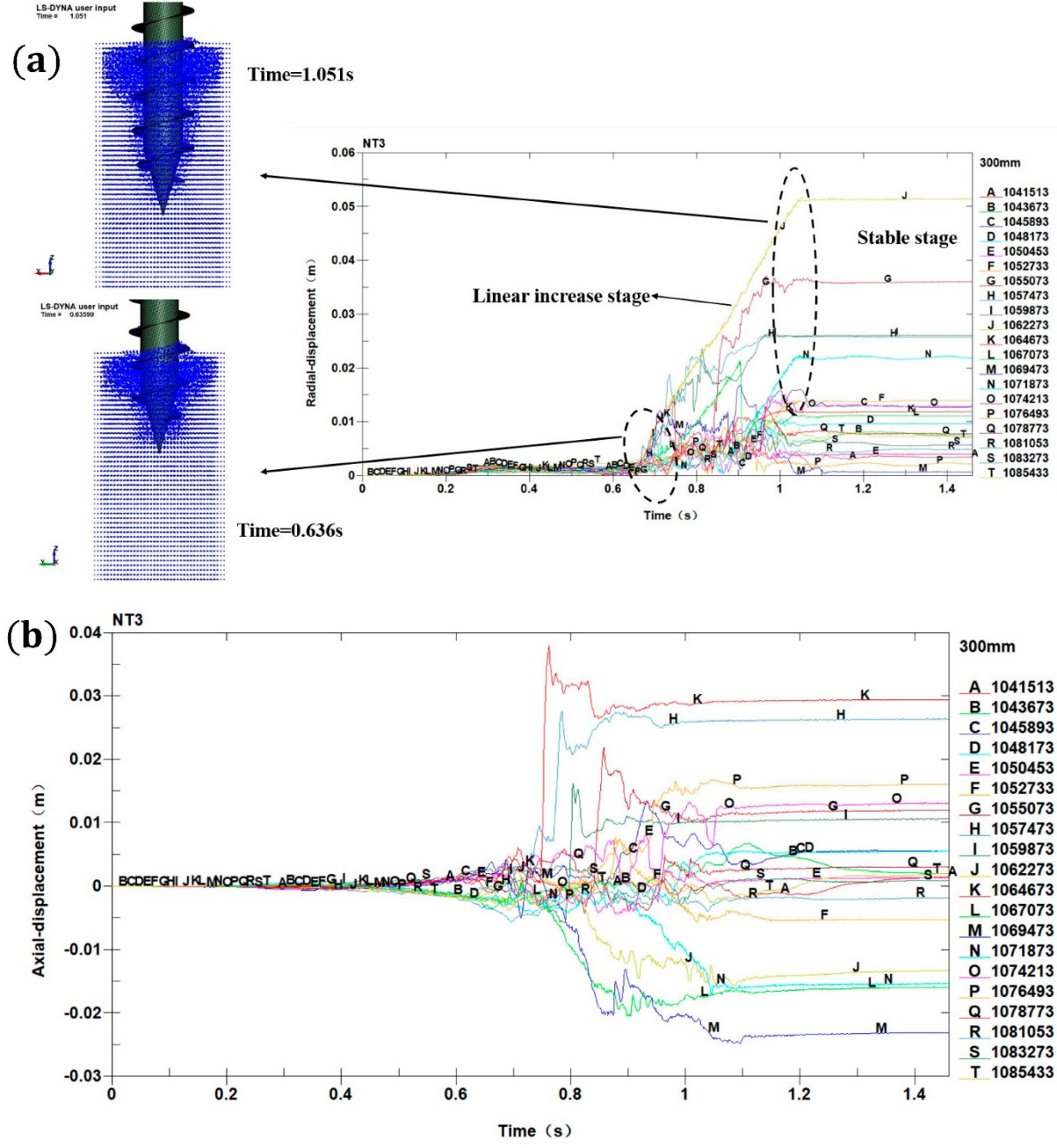

**Figure 15.** Displacement of the middle soil particles in different directions: (**a**) radial and (**b**) axial.

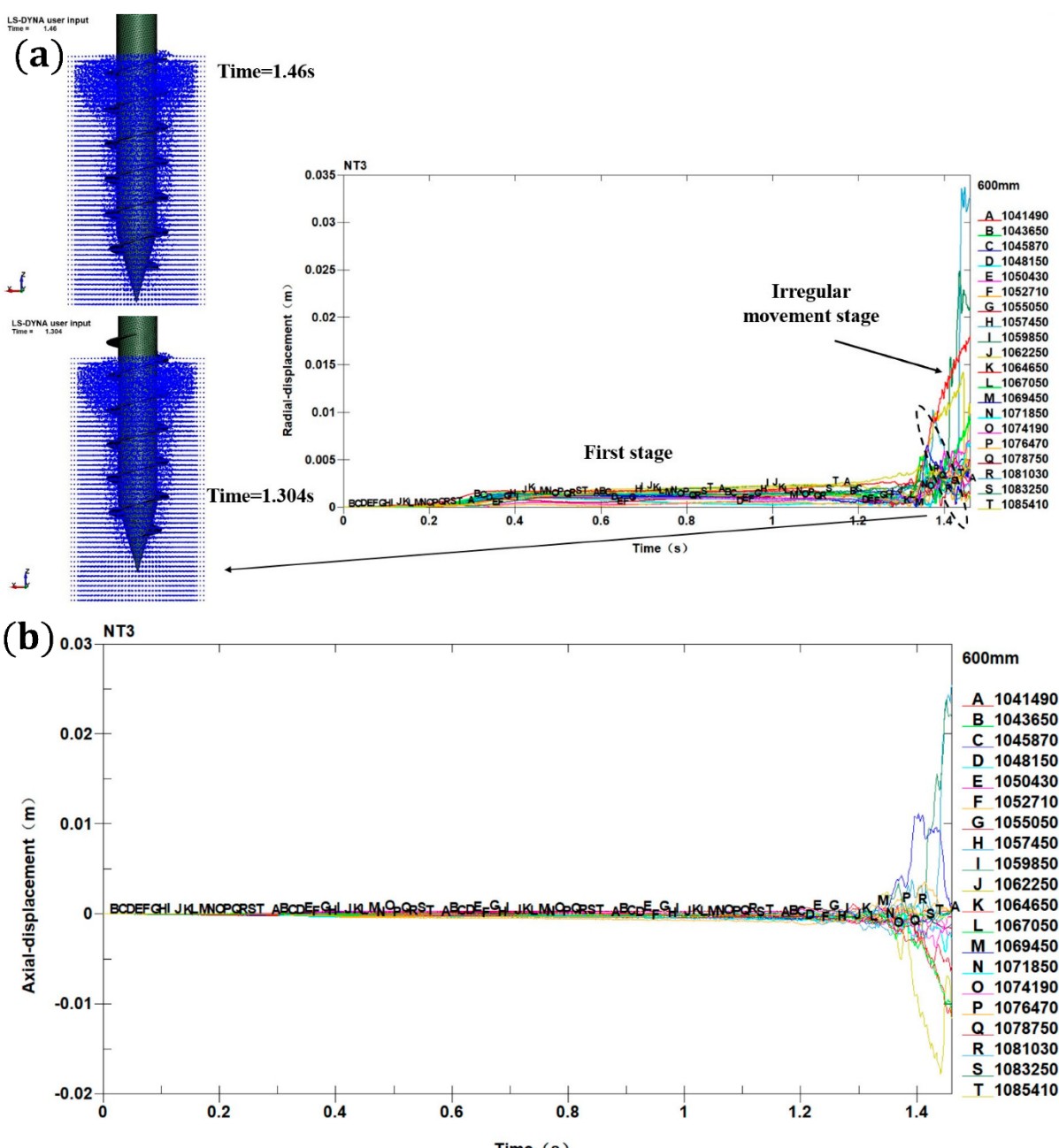

**Figure 16.** Displacement of the deep soil particles in different directions: (**a**) radial; (**b**) axial.

### 4.2. Influence of Pile Structure Parameters on Soil Displacement

Figure 17 shows the soil displacement contours at the end of the installation of different screw piles. It can be found that there is similarity in the range of soil disturbance caused by the installation of different screw piles, and the soil uplift phenomenon at the surface can be observed in all cases. The installation of all screw piles exhibits a greater range of disturbance to the shallow soil. This phenomenon is consistent with the previous analysis. In the installation process, with the installation depth increase, the extent of soil disturbance caused by the screw pile will decrease. Compared to CSFM, there is a greater damage range to the soil by HFM, indicating that HFM has a greater impact on soil disturbance than CSFM during the installation process.

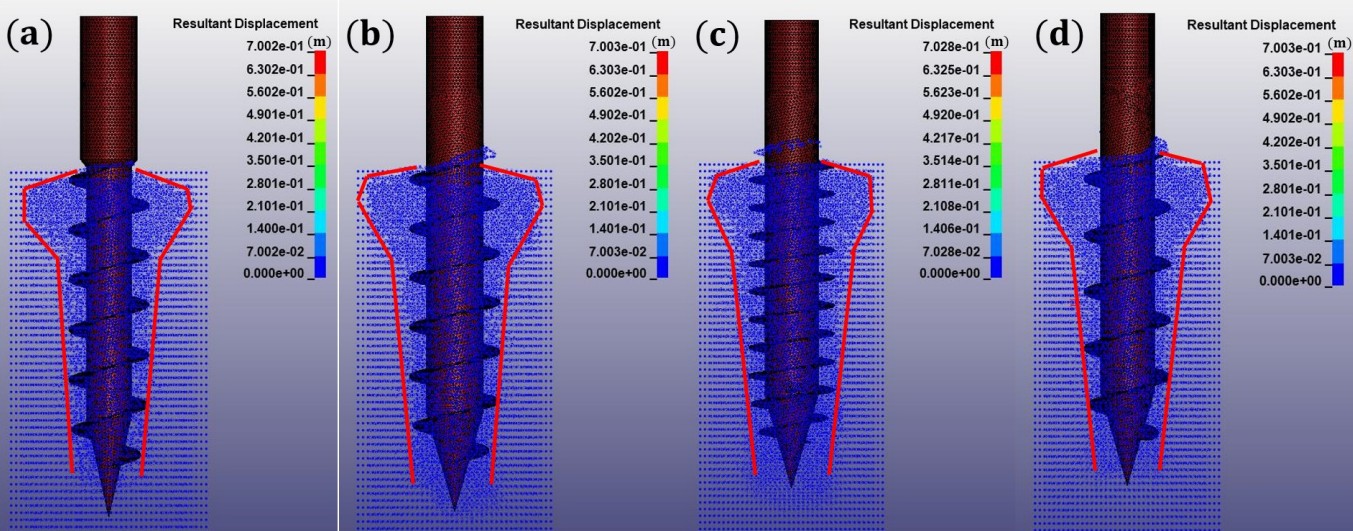

**Figure 17.** Soil displacement contour caused by the installation of the screw pile: (**a**) NT2, (**b**) NT3, (**c**) NT4 and (**d**) NT5.

For NT2 and NT3 (Figure 17a,b), they have different pile diameters. As the pile diameter increases, the friction between the screw pile and the surrounding soil will increase. Since the rotation of the screw pile, the larger friction will lead to the movement of more soil particles, causing a larger damage surface between the soils. It can also be seen from Figure 17b that the installation of NT3 exhibits a greater range of damage to the soil. The different ranges of soil disturbance for the two piles indicate that the variation in pile diameter will influence the range of soil damage by CSFM and HFM. For NT4 and NT5 (Figure 17c,d), it can be seen that the damage extent of the shallow soils of the two piles shows a consistency. However, in the case of larger spiral pitch, the damage range of the middle and deep soils is smaller and the soil damage range is more closely fitted to the inter-helical profile. It indicates that when the spiral pitch is changed, the disturbance range by HFM on the surface soil is not affected, and only the disturbance range by CSFM on the middle and deep soil is changed. This means that the change in the area of soil damage by HFM is not affected by the variation in the spiral pitch. For this micro screw pile, when the spiral pitch decreases, it will make the number of blades increase due to the invariable total height of the helical blades. Then the closed-arranged blades will produce more frequent shearing action on the soil, and the surrounding soil around the pile will be broken up. Therefore, this causes greater displacement of soil particles in the radial and axial directions, which also exhibits a greater damage range of the soil.

### 4.3. Influence of Pile Structure Parameters on Soil Pressure

As shown in Figure 18, it can be seen that there is consistency in the range of soil pressure variation during the installation of different screw piles. The obvious areas of soil pressure variations are mainly concentrated near the pile body, the helical blades and the conical pile body, indicating that it is these three parts that interact with the soil during the installation. For these micro screw piles with continuous blades, the installation force of the screw piles is mainly generated by the force between the soil and spiral blades. Therefore, as shown in Figure 18, most of the soil particles with higher pressure are located between the spiral blades. It can be seen that the shallow soil pressure is lower than that of the middle and deep soil at the end of the installation of the screw pile. Considering the soil failure mechanism of the screw pile during the installation process, it shows that the effect of HFM on the surrounding soil pressure around the screw pile is weaker than that of CSFM, and the variation in soil pressure is mainly caused by CSFM.

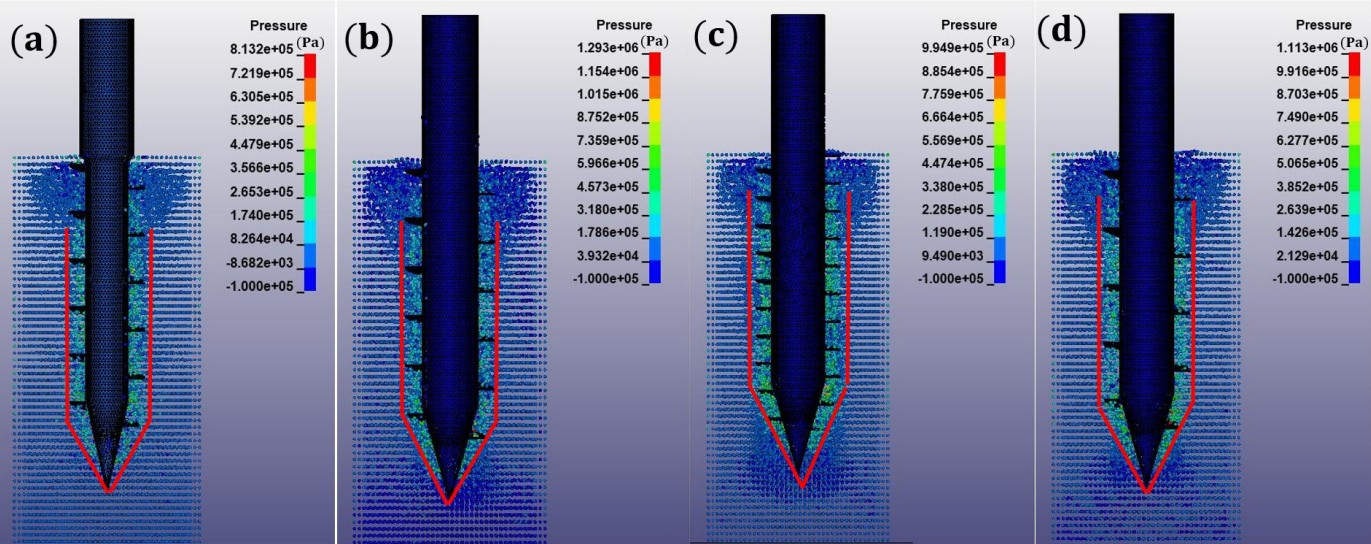

**Figure 18.** Soil pressure contour caused by the installation of the screw pile: (**a**) NT2, (**b**) NT3, (**c**) NT4 and (**d**) NT5.

For NT2 and NT3, as is shown in Figure 18a,b, the max soil pressure around NT3 is greater than that of NT2, which is $1.293 \times 10^6$ Pa. It can be concluded that the maximum pressure of soil particles increases with the increase in pile diameter during the installation process. We can see from Figure 18a,b that the variation range of soil pressure is also related to pile diameter, which is positively correlated with pile diameter. For the two screw piles with different pile diameters, both the soil pressure values and the range of soil pressure variation around the pile are also different. This shows that when the pile diameter is varied, the two soil failure mechanisms caused by the installation of the screw pile will change the range of soil pressure around the pile and soil pressure values. This is mainly because, with the increase in the pile diameter, a larger contact area is created between the screw pile and the soil, resulting in a stronger squeezing effect and a higher pressure on the soil around the pile.

For NT4 and NT5 (Figure 18c,d), the soil around the NT5 exhibited a greater pressure at the end of installation, which is $1.113 \times 10^6$ Pa. The two screw piles have different spiral pitch, indicating the maximum soil pressure around pile increases with the increase in pitch during the installation process. It can be seen from Figure 18c,d that with the decrease in the pitch, the range of soil pressure around the pile does not vary, but the maximum value of soil pressure becomes smaller. From the above analysis, it can be seen that when the pitch is varied, the two soil failure mechanisms caused by the installation of the screw pile do not change the range of soil pressures around the pile, but the soil pressure values will be changed. This is because the closed-arranged blades will have a stronger shearing action with the soil particles around the pile, and the surrounding soil will be broken up. Therefore, the interlocking forces between the soil particles will become smaller, resulting in a greater range of pressure variations and a smaller pressure on the soil around the pile.

### 4.4. Influence of Pile Structure Parameters on Soil Stress

During the installation process, due to the squeezing and shearing of the screw pile on the surrounding soil, the soil stress variations around the screw pile can be observed significantly. At the end of the installation of the screw pile, the state of soil shear stresses around the pile is shown in Figure 19. For different screw piles, there is a consistency in the variation range of the surrounding soil stress, and the stress area is approximately symmetrical. The maximum soil stress values around all different piles are $2.309 \times 10^5$ Pa, and the areas of soil stress variation are all located near the screw piles.

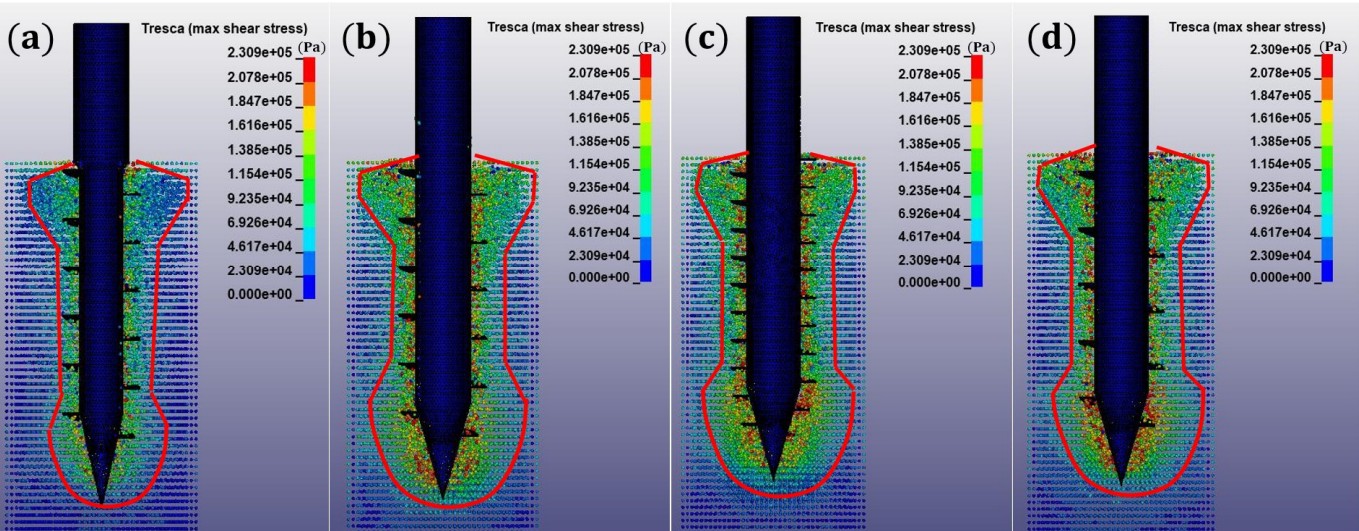

**Figure 19.** Soil stress contour caused by the installation of the screw pile: (**a**) NT2, (**b**) NT3, (**c**) NT4 and (**d**) NT5.

As is shown in Figure 19, for different screw piles, all the variation areas of soil stress develop into different shapes at different depths, which is related to the soil failure mechanism caused by the installation of the screw pile. For the shallow soil, the range of stress variation develops into a funnel shape, and the size of this zone is related to the pile's structural parameters. It can be seen that for the four screw piles, the variation range of shallow soil stress around NT4 is almost equal to that of NT5 (Figure 19c,d), while the stress variation range of NT3 is greater than that of NT2 (Figure 19a,b), but none of the stress variations extend to the soil boundary. The damage to shallow soils is mainly caused by HFM, indicating that the range of stress variation by HFM on the shallow soil is only related to the pile diameter, and it is irrelevant to the spiral pitch. The larger the pile diameter, the greater the range of stress variation in the shallow soil.

For the middle soil, the range of stress variation corresponds to the inter-helical profile, which is approximated as the size of the screw diameter. It can be found from Figure 19c,d that the stress range of NT4 is larger than that of NT5, indicating that the variation in spiral pitch will change the stress variation range of the middle soil caused by CSFM, which will increase with the decrease in spiral pitch. This is mainly related to the helical blade's shearing action. For NT2 and NT3 (Figure 19a,b), the soil stress range of NT2 is smaller than that of NT3. The pile diameters of the two screw piles are different, but all other parameters are the same, indicating that the variation in pile diameter will change the stress variation range of the middle soil caused by CSFM, which will increase with the increase in pile diameter. This is related to the squeezing effect of the screw pile. As is shown in Figure 19, it can be seen that for the range of stress variation at different depths, the stress range in the shallow soil is greater than that in the middle soil, indicating that HFM has a stronger effect on the soil stress around the screw pile than that of CSFM during the installation process.

For the deep soil, as is shown in Figure 19, it can be seen that for the different screw piles, the soil stress variation around the pile develops into a circular region, the size of which is approximately twice the pile diameter. As is presented in Figure 19, the circular variation range of deep soil stress around NT4 is almost equal to that of NT5, while NT3 has a greater range of stress variation than NT2, indicating that the stress region around the pile in the deep soil is related to the pile diameter and unrelated to the spiral pitch. This phenomenon is mainly related to the resistance of the pile tip during the installation process. For the range of soil stress variation around the pile tip, it can reflect the magnitude of the tip resistance. During the installation process, the tip resistance of is usually correlated with the taper angle of the pile tip. The greater the taper angle, the greater the end resistance

of the pile. As is shown in Table 2, for NT4 and NT5, the pile diameters of them are the same, which means the pile tip is subjected to equal resistances, thus also resulting in an approximately equal stress plastic variation circular region of them. When the pile diameter is different, the pile tip resistance is positively related to the diameter. Therefore, the circular stress variation area of NT3 is greater than that of NT2 (Figure 19a,b). Considering the soil failure mechanisms caused by the installation of the screw pile, it can be seen from Figure 19 that for the range of stress variation at different depths, the stress range in the deep soil is greater than that in the middle soil, indicating that CSFM has a stronger effect on the soil stress around the screw pile in deep soil than that in the medium soil during the installation process. However, the disturbance displacement of soil particles in the middle layer is greater than that in the deep layer, combined with the previous analysis, which also precisely shows that there is a stronger force interaction between the pile tip and the deep soil. However, because of the action of the soil's overburden pressure, this again results in a minimal disturbance displacement of the deep soil particles.

## 5. Conclusions

In this paper, a FEM-SPH coupling model was developed to analyze the large deformation problem caused by the installation of a screw pile. The screw pile was modelled using the finite element method and the soil model was built using the SPH method; the installation process of the screw pile is well simulated. Further, the FEM-SPH simulation results were compared with the field tests' results and traditional finite element simulation result. In addition, based on the established FEM-SPH numerical model, the differences in different screw piles were analyzed during the installation process. Based on the analyses conducted in this paper, the following conclusions can be obtained:

(1) By comparing with the experimental results, the accuracy of the FEM-SPH coupling model is verified. FEM-SPH simulation results show a better agreement with the experimental results in terms of the installation torque and the soil deformation of the ground surface, indicating that the FEM-SPH method has good applicability in dealing with the large deformation caused by the installation of the screw pile.

(2) Compared to the traditional FE (AlE) method, the FEM-SPH method possesses better efficiency and accuracy at the same time, which is more suitable for simulating the large deformation problem caused by the installation of the screw pile.

(3) The FEM-SPH numerical model was used to analyze the installation process of the screw pile. The numerical results show that the interaction during the installation process between the screw pile and the surrounding soil is transferable, and the load transfer occurs throughout the installation of the screw pile. With the increase in the installation depth, the extent of damage to the soil caused by the screw pile will decrease. The installation of the screw pile results in heave damage to the shallow soil and exhibits cylindrical shear damage to the middle and deeper layers.

(4) Based on the developed numerical model, the variability in the installation process of different screw piles was analyzed. The analysis shows that the two failure mechanisms (HFM and CSFM) have different effects on the soil during the installation of the screw pile. For HFM and CSFM caused by the installation of the screw pile, the influence range on the soil around the pile of HFM is only related to the pile diameter and is irrelevant to the spiral pitch. Unlike HFM, the influence range on the soil around the pile of CSFM is affected by the variations in both spiral pitch and pile diameter.

**Author Contributions:** Conceptualization, Q.Z.; Writing—original draft, Q.Z.; Formal analysis, Q.Z.; validation, Q.Z.; Project administration, Y.W. and Y.T.; Writing—review and editing, Q.Z. and Y.W.; Supervision, Y.W. and Y.T.; Data curation, Y.W. and Y.T.; Investigation G.R. and Z.Q.; Visualization Z.Q. and W.L.; Software, Z.Y. and W.L. All authors have read and agreed to the published version of the manuscript.

**Funding:** The authors highly appreciate the financial support of this study by Guangdong Water Conservancy Science and Technology Innovation Fund project (No. 2017-31).

**Institutional Review Board Statement:** Not applicable.

**Informed Consent Statement:** Not applicable.

**Data Availability Statement:** Data sharing is not applicable.

**Conflicts of Interest:** The authors declare no conflict of interest.

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
