# Peer review of "Numerical Analysis of the Installation Process of Screw Piles Based on the FEM-SPH Coupling Method"

_applsci, doi:10.3390/app12178508_

Round 1

Reviewer 1 Report

Due to the rapid publication cycle of MDPI, I must recommend rejection of this paper. This paper has a potential to be greatly improved and becoming worthy of publication in the journal with an Impact Factor of 2.8. However, the issues that need to be resolved would require a few months of research/modelling/analysis and cannot be fixed in a matter of days. I recommend to authors to thoroughly revise the paper and to resubmit it to the same journal once the issues have been fixed.

The first thing I must point out, is that the units through the whole paper are inconsistent. This makes the readers question the validity of the results, take Table 1 and Table 3 as an example. The Young's modulus (E) for steel is 2.1 e+05 MPa, while Table 3 says it is 21 !!!!! This is probably a typo error, but it may be an error in the LS-Dyna input file

I recommend to authors to use the table available in LS-Dyna Support at https://www.dynasupport.com/howtos/general/consistent-units

Now, the main thing authors need to address in this paper is significance. There is no new numerical method in this paper, no new contact algorithm, no new material model, nothing new. In fact, the authors use existing, commercial LS-Dyna software to simulate installation of screw piles into soil. So they need to prove that the results they showed could be beneficial to someone. For example, how can a civil engineer use their work to select the most appropriate screw pile?

Since a commercial LS-Dyna software is used, a theoretical background given in section 2 is oversized, Figures 1,2 and 3 are not necessary, but instead, authors should describe the most important LS-Dyna keywords that they used (values of parameters and their meaning). For example, there is a sentence “The contact between the screw pile and the soil is a point-surface contact, where the SPH particles of the soil model are slave contact points and the FE nodes of the pile model are master contact surfaces.”, so I assume AUTOMATIC_NODES_TO_SURFACE keyword was used in which case, what were the values of Static coefficient of friction (FS) and Dynamic coefficient of friction (FD)?

It appears that the authors used 001-ELASTIC for steel screw pile and 005-SOIL_AND_FOAM material model for soil. Perhaps 025-GEOLOGIC_CAP_MODEL would be more appropriate for the soil, perhaps not, but it is worth investigating.

From the Figure 10 readers can see that the authors used Arbitrary Lagrangian-Eulerian Formulation (ALE) for FEM modelling, and only from this figure, there is not a single word in text about this methodology (and of course LS-Dyna keywords used to define it).

The main reason for my recommendation for the rejection of this paper is Figure 9 d) and to a lesser extent Fig 9 b), Fig 11 b) and figures 16,17 and 18. According to the part of the Archimedes’ principle: The volume of displaced fluid is equivalent to the volume of an object fully immersed in a fluid or to that fraction of the volume below the surface for an object partially submerged in a liquid. I know the soil is not water, and that is compressible, but if you look at the Figure 9 a) you can see how much of the soil volume should a screw pile expel from the ground.

In the case of the FEM model, I assume there is an erosion of the soil elements, which could also explain the prolonged computational time as well. Everyone who ever had a misfortune of using the SPH method in their analysis knows how computationally demanding this method is, how sensitive it is, and prone to instability. FEM is superior to SPH in every way in 99.9% of the times, but in the case of screw pile modelling, coupling of SPH with FEM is the best, and perhaps the only choice.

I would recommend to authors to try to use ERODING_SURFACE_TO_SURFACE or ERODING_NODES_TO_SURFACE keyword for a contact definition. Also to check all the other options in LS-Dyna concerning element erosion. Perhaps there is something to be done in the ALE keywords as well, in order to get more realistic results.

Figure 9. a) cannot be presented as a valid scientific/engineering result, it can only show why FEM-SPH coupling is required

As for the SPH results, they look better, but still the amount of expelled soil is unrealistically small. In fact, it appears as if the only expelled soil lies on top of the spiral blade.

In this case, I would recommend to the authors to try to use a constant smoothing length for SPH particles (defined by HMIN=1 and HMAX=1 values in the SECTION_SPH keyword. Also 025-GEOLOGIC_CAP_MODEL material model may yield better results. Or it may not, either way it’s worth investigating and the results, whatever they might be, would still represent a scientific contribution.

Another thing that should be investigated is mesh sensitivity analysis, or in case of the SPH, particle density analysis. The authors used particle spacing of 0.01 mm, what would the results look like if they used for example 0.011mm or 0.009mm. The same goes for the FEM element size, Authors should try to increase or decrease the element size until the results convergence is achieved.

In the Figure 14 the authors report violent motion of soil particles. Although violent motion is realistic in the microscopic scale, where force chains between real granular particles are broken (which can be simulated using Discrete Element Method), on the macroscopic scale SPH pseudo-particles represent sections of modelled continuum (same as the finite elements), hence one would not expect anything sudden or violent. It is more likely this behavior is the result of some numerical instability, probably caused by the contact parameters.

Sections 4.1.1, 4.1.2, and 4.1.3 do not show anything significant, perhaps if the shallow, middle and the deep soil were modelled with different material (or the same material with different parameters) there could be something worth reporting, but in the present form, it’s just a detailed description of soil behavior which everyone would intuitively expect.

Sections 4.2, 4.3, and 4,4 and figures 16,17 and 18 are interesting and could justify the publication of this paper, but authors should first try to improve the amount of the expelled soil as mentioned before, and second, they should describe real world applications of their findings (as I also pointed out at the beginning)

Author Response

Dear reviewers:

Thank you very much for your kindly comments on our manuscript. There is no doubt that these comments are valuable and very helpful for revising and improving our manuscript. We have studied comments carefully and have made revision which marked in red in revised paper. We have tried our best to revise our manuscript according to the comments. Attached please find the revised version, which we would like to submit for your kind consideration.

Responses to the reviewer’s comments

Reviewer #1:

  1. Response to comment:

Due to the rapid publication cycle of MDPI, I must recommend rejection of this paper. This paper has a potential to be greatly improved and becoming worthy of publication in the journal with an Impact Factor of 2.8. However, the issues that need to be resolved would require a few months of research/modelling/analysis and cannot be fixed in a matter of days. I recommend to authors to thoroughly revise the paper and to resubmit it to the same journal once the issues have been fixed.

Response:

  First of all, thank you for carefully reviewing our manuscript and giving suggestions. we would like to answer the questions you mentioned and give detailed account of the changes made to the original manuscript. The main corrections in the paper and the responses to the reviewer’s comments are as following, we would like to submit for your kind consideration.

  1. Response to comment:

The first thing I must point out, is that the units through the whole paper are inconsistent. This makes the readers question the validity of the results, take Table 1 and Table 3 as an example. The Young's modulus (E) for steel is 2.1 e+05 MPa, while Table 3 says it is 21 !!!!! This is probably a typo error, but it may be an error in the LS-Dyna input file.

I recommend to authors to use the table available in LS-Dyna Support at https://www.dynasupport.com/howtos/general/consistent-units.

Response:

  We thank you for your reminding us this important point. According to your suggestion, we carefully checked the units used in the numerical simulation. In our numerical simulation, a unified unit system (kg,m,pa,J) has been adopted. For the problem of the Young's modulus (E) in Table 3, this is a typo error and has been revised.

  1. Response to comment:

Now, the main thing authors need to address in this paper is significance. There is no new numerical method in this paper, no new contact algorithm, no new material model, nothing new. In fact, the authors use existing, commercial LS-Dyna software to simulate installation of screw piles into soil. So they need to prove that the results they showed could be beneficial to someone. For example, how can a civil engineer use their work to select the most appropriate screw pile?

Response:

We gratefully appreciate for your valuable suggestion. In the current research, the analysis of screw pile mostly focuses on the bearing process (Livneh et al. 2008; Jabur et al. 2021; Khan and Siddiqua 2018; Harnish and Naggar 2017; Giovanni et al. 2018). For the research of bearing capacity in most cases, researchers rarely consider the installation of pile. In fact, the installation of screw pile will have a certain impact on the bearing capacity, and the installation of screw pile is a dynamic large deformation process. Therefore, the bearing capacity of pile may be underestimated or overestimated in these cases. In this paper, the FEM-SPH method was used to analyze the installation process of screw pile. For example, according to our analysis, the installation torque of screw pile is related to the pile structure parameters, and the larger the installation torque, the greater the impact range of screw pile on the soil. In their studies (Livneh et al. 2008; Jabur et al. 2021; Khan and Siddiqua 2018; Harnish and Naggar 2017; Giovanni et al. 2018), the installation torque is directly related to the pile bearing capacity. Therefore, the bearing capacity of screw pile may be able to related with the pile structural parameters and the soil change range during the installation process by other researchers. Then, the bearing capacity of different piles can be compared by the damage range of soil during the installation process, which can save the research cost. The installation of screw pile can only be completed with torque provided by special equipment. However, if the torque is too small, the pile cannot be installed, and excessive torque will destroy the screw pile structure. In this paper, the accuracy of FEM-SPH method in predicting torque is proved by comparing the experimental and simulation results. For a civil engineer, when installing some uncertain screw piles, the installation torque can be preliminarily predicted through numerical simulation, so as to select appropriate equipment to reduce losses.

Livneh B, El Naggar MH (2008) Axial testing and numerical modeling of square shaft helical piles under compressive and tensile loading. Can Geotech J 45(8):1142–115

Jabur J, Mahmood MR, Abbas SF (2021) The effect of installation torque on the behavior of helical piles under compression )load tested by centrifuge device. IOP Conference Series: Mater Sci Eng 94(1):01209

Khan U, Siddiqua S (2018) Study of compressive loading capacities of helical piles using torque method and induced settlements. Environ Earth Sci 77(1):22–33

Harnish J, El Naggar MH (2017) Large diameter helical pile capacity torque correlations. Can Geotech J 54(7):968–986

Giovanni S, Cristina DHCT, Pierpaolo O, Carlos MMS (2018) Estimation of uplift capacity and installation power of helical piles in sand for offshore structures. J Waterw Port Coast Ocean Eng

144(6):04018019

  1. Response to comment:

Since a commercial LS-Dyna software is used, a theoretical background given in section 2 is oversized, Figures 1,2 and 3 are not necessary, but instead, authors should describe the most important LS-Dyna keywords that they used (values of parameters and their meaning). For example, there is a sentence “The contact between the screw pile and the soil is a point-surface contact, where the SPH particles of the soil model are slave contact points and the FE nodes of the pile model are master contact surfaces.”, so I assume AUTOMATIC_NODES_TO_SURFACE keyword was used in which case, what were the values of Static coefficient of friction (FS) and Dynamic coefficient of friction (FD)?

Response:

We gratefully appreciate for your valuable suggestion. we mainly want to explain the calculation process of the installation torque of screw pile in the numerical simulation from the microscopic perspective through the relationship between SPH particles and element nodes. For Fig.1 and Fig.2, it is mainly to facilitate readers who are not familiar with the FEM-SPH method to better understand the relationship between SPH particles and the relationship between SPH particles and element nodes. For Fig.3, the overall content of Section 3 is summarized to make it easier for readers to read.

According to your suggestions, the important keywords used in the numerical simulation have been added in the paper, and the specific additions can be viewed in the revised manuscript(Fig.5). The soil in Guangzhou is lateritic soil with a friction coefficient of 0.2-0.4. Therefore, for the contact keywords used in the FEM-SPH model, in the numerical simulation, the value of Static coefficient of friction (FS) was used 0.4 and the value of Dynamic coefficient of friction (FD) was used 0.3. For the FEM numerical model (ALE), the friction factor can be added in the keyword CONSTRAINED_ LAGANGE_ IN_ SOILD, the value was used 0.3. For the value of the friction coefficient, this is also one of the reasons for the error between the test results and the simulation results.

  1. Response to comment:

It appears that the authors used 001-ELASTIC for steel screw pile and 005-SOIL_AND_FOAM material model for soil. Perhaps 025-GEOLOGIC_CAP_MODEL would be more appropriate for the soil, perhaps not, but it is worth investigating.

Response:

We gratefully appreciate for your valuable suggestion. As for the establishment of the numerical model, considering the small deformation of the screw pile. 020-RIGID model was used for screw pile because it mainly studies the variation of the soil during the installation process. For the soil  material model, 005-SOIL_AND_FOAM and 147-SOIL_FHWA_SOIL was used respectively in the earlier study. By comparing with the test results, we find that 005-SOIL_AND_FOAM material model seems to be more suitable for the soil model, at least for this paper. For your suggestions(025-GEOLOGIC_CAP_MODEL was used for the soil), we will further explore and study this model, and hope that the future work can be further sent to you for review.

  1. Response to comment:

From the Figure 10 readers can see that the authors used Arbitrary Lagrangian-Eulerian Formulation (ALE) for FEM modelling, and only from this figure, there is not a single word in text about this methodology (and of course LS-Dyna keywords used to define it).

Response:

We thank you for your reminding us this important point. According to your suggestions, the important keywords used by FEM model (ALE) was added in section 3.1, and the specific additions can be viewed in the revised manuscript(Fig.5 in the revised manuscript).

  1. Response to comment:

The main reason for my recommendation for the rejection of this paper is Figure 9 d) and to a lesser extent Fig 9 b), Fig 11 b) and figures 16,17 and 18. According to the part of the Archimedes’ principle: The volume of displaced fluid is equivalent to the volume of an object fully immersed in a fluid or to that fraction of the volume below the surface for an object partially submerged in a liquid. I know the soil is not water, and that is compressible, but if you look at the Figure 9 a) you can see how much of the soil volume should a screw pile expel from the ground.

Response:

We gratefully appreciate for your valuable suggestion. According to your suggestions, for the FEM model, other options in LS-Dyna concerning element erosion was checked and modified. As shown in Fig.10(d) in the revised version, a more realistic effect was obtained.

  1. Response to comment:

In the case of the FEM model, I assume there is an erosion of the soil elements, which could also explain the prolonged computational time as well. Everyone who ever had a misfortune of using the SPH method in their analysis knows how computationally demanding this method is, how sensitive it is, and prone to instability. FEM is superior to SPH in every way in 99.9% of the times, but in the case of screw pile modelling, coupling of SPH with FEM is the best, and perhaps the only choice.

Response:

We gratefully appreciate for your valuable suggestion. As for the simulation calculation time, as you said, in the FE method(ALE), due to the penetration of the spiral pile, there will has an erosion of the soil elements, making the contact between screw pile and the soil more complicated, thus lengthening the simulation calculation time. In addition, the FEM simulation(ALE) may fail due to excessive deformation in the process of grid erosion. The SPH method has good applicability for large deformation problems, in the simulation process, the soil was modelled by SPH method, which can well avoid the wrong termination problem caused by large mesh deformation in the FEM method(ALE). In addition, for the established FEM-SPH model, since the nodes of the finite element mesh of screw pile have little or no deformation in the calculation process, the contact calculation between the element nodes and SPH particles is easier than that in the ALE method. This is also an important reason that the calculation time of FEM-SPH model is less than that of ALE method.

  1. Response to comment:

I would recommend to authors to try to use ERODING_SURFACE_TO_SURFACE or ERODING_NODES_TO_SURFACE keyword for a contact definition. Also to check all the other options in LS-Dyna concerning element erosion. Perhaps there is something to be done in the ALE keywords as well, in order to get more realistic results.

Response:

We gratefully appreciate for your valuable suggestion. According to your suggestions, ERODING_NODES_TO_SURFACE was used to define the contact between screw pile and the soil. By comparing with the simulation results of the AUTOMATIC_NODES_TO_SURFACE keyword, it is found that there is little difference between the two results, so it is considered that AUTOMATIC_NODES_TO_SURFACE keyword is feasible for the pile-soil contact in this paper.

Considering you suggestion, we will do further study on ERODING_NODES_TO_SURFACE keyword to improve the simulation results. According to your suggestions, for the FEM model, other options in LS-Dyna concerning element erosion was checked and modified. As shown in Fig.10(d) in the revised version, a more realistic effect was obtained.

  1. Response to comment:

Figure 9. a) cannot be presented as a valid scientific/engineering result, it can only show why FEM-SPH coupling is required。 As for the SPH results, they look better, but still the amount of expelled soil is unrealistically small. In fact, it appears as if the only expelled soil lies on top of the spiral blade. In this case, I would recommend to the authors to try to use a constant smoothing length for SPH particles (defined by HMIN=1 and HMAX=1 values in the SECTION_SPH keyword. Also 025-GEOLOGIC_CAP_MODEL material model may yield better results. Or it may not, either way it’s worth investigating and the results, whatever they might be, would still represent a scientific contribution.

Response:

  We thank you for your reminding us this important point. For Fig.9(a)(Fig.10(a) in the revised manuscript), we take it as a supplementary verification of the simulation results with FEM-SPH method and FEM method(ALE). For the numerical model, it is verified by comparing with the installation torque of screw pile measured by the field test. In terms of the amount of the expelled soil, we feel sorry for the difference between the results of SPH and the experiment. In our current simulation, we have used a constant smooth length for SPH particles. We are also adjusting relevant keywords and parameters according to your suggestions. We hope that the simulation can be more realistic. Therefore, we also hope that future work can still be sent to you for review in the hope of getting better suggestions.

  1. Response to comment:

Another thing that should be investigated is mesh sensitivity analysis, or in case of the SPH, particle density analysis. The authors used particle spacing of 0.01 mm, what would the results look like if they used for example 0.011mm or 0.009mm. The same goes for the FEM element size, Authors should try to increase or decrease the element size until the results convergence is achieved.

Response:

We thank you for your reminding us this important point. Mesh sensitivity analysis is indeed a matter of concern. In fact, when we initially adopted this method, a larger(0.02mm) and smaller(0.001mm) particle spacing were used for numerical simulation. In the case of larger particle spacing, the amount of expelled soil is smaller. However, in the case of smaller particle spacing, the simulation calculation time will become very long. In combination with Wang's study (Wang et al. 2021), so the particle spacing (0.01mm) was used to establish the soil model. In addition, in order to compare the applicability of FEM-SPH and FEM(ALE) methods. The particle spacing is the same size as the finite element mesh.

Wang W, Wu Y.J., Wu H., Yang C.Z., Feng Q.S. Numerical analysis of dynamic compaction using FEM-SPH coupling method. Soil Dynamics and Earthquake Engineering. 2021, 140, 106420

  1. Response to comment:

In the Figure 14 the authors report violent motion of soil particles. Although violent motion is realistic in the microscopic scale, where force chains between real granular particles are broken (which can be simulated using Discrete Element Method), on the macroscopic scale SPH pseudo-particles represent sections of modelled continuum (same as the finite elements), hence one would not expect anything sudden or violent. It is more likely this behavior is the result of some numerical instability, probably caused by the contact parameters.

Response:

We thank you for your reminding us this important point. For the violent movement of soil particles shown in Fig.14, this is mainly caused by the penetration of the conical pile head of screw pile. As mentioned above, the installation of screw pile will discharge the soil, which will cause a large displacement of this part of the soil. This is also the main reason for the violent movement of soil particles around the pile. It can be seen from Fig.14 (Fig.15 in the revised manuscript) that the violent movement of particles is not always carried out, but tends to be stable after a period of time(after the installation of pile tip), indicating that the possibility of numerical instability is very small. For your suggestions (It is more likely this behavior is the result of some numerical instability, probably caused by the contact parameters), we will also consider it carefully in the follow-up research and analysis.

  1. Response to comment:

Sections 4.1.1, 4.1.2, and 4.1.3 do not show anything significant, perhaps if the shallow, middle and the deep soil were modelled with different material (or the same material with different parameters) there could be something worth reporting, but in the present form, it’s just a detailed description of soil behavior which everyone would intuitively expect.

Response:

We gratefully appreciate for your valuable suggestion. For sections 4.1.1, 4.1.2 and 4.1.3, the movement of soil particles at three depths was described, which is of great significance to understand the variation of the soil at different depths during the installation process of screw pile. According to our analysis, it can be seen that the movement law of soil particles at different depths is different. For this kind of micro screw pile, the installation of it is usually be completed with torque provided by low power equipment. In practical use, in order to make it easier to install, the front end of the pile usually has this tapered structure. In this paper, according to our analysis, the movement of soil particles is mainly caused by the installation of the tapered pile head of this type of screw pile. Therefore, combined our analysis, changing the pile head structure may reduce the damage to the soil and the torque required for the installation, which is of great significance for improving the efficient installation of screw piles. Of course, we are also very grateful for your suggestion: new soil models and different materials (or the same materials and different parameters) are used to simulate the soil. We are also conducting new research and analysis on this, so as to obtain new analysis results from a new perspective.

  1. Response to comment:

Sections 4.2, 4.3, and 4,4 and figures 16,17 and 18 are interesting and could justify the publication of this paper, but authors should first try to improve the amount of the expelled soil as mentioned before, and second, they should describe real world applications of their findings (as I also pointed out at the beginning)

Response:

We gratefully appreciate for your valuable suggestion. According to your suggestion, some keywords has been adjusted in FEM method (ALE), and the amount of the expelled soil was indeed improved (Fig.10 in the revised manuscript). As for the simulation results of SPH method, we will continue to make further adjustments according to your suggestions, with a view to further improving the amount of the expelled soil.

In this paper, the FEM-SPH method was used to analyze the installation process of screw pile. For example, according to our analysis, the installation torque of the screw pile is related to the pile structure parameters, and the larger the installation torque, the greater the impact range of screw pile on the soil. In their study, the installation torque is directly related to the pile bearing capacity. Therefore, the bearing capacity of screw pile may be able to related with the pile structural parameters and the soil change range during the installation process by other researchers. Then, the bearing capacity of different piles can be compared by the damage range of soil during the installation process, which can save the research cost. The installation of the pile can only be completed with torque provided by special equipment. However, if the torque is too small, the pile cannot be installed, and excessive torque will destroy the screw pile structure. In this paper, the accuracy of FEM-SPH method in predicting torque is proved by comparing the experimental and simulation results. For a civil engineer, when installing some uncertain screw piles, the installation torque can be preliminarily predicted through numerical simulation, so as to select appropriate equipment to reduce losses.

In addition, the installation process of screw pile is analyzed by the motion law of soil particles at different depths. the movement of soil particles is mainly caused by the installation of the tapered pile head of this type of screw pile. Therefore, combined our analysis, changing the pile head structure may reduce the damage to the soil and the torque required for the installation, which is of great significance for improving the efficient installation of screw piles.

Finally, thanks again for your valuable comments, which are very important and instructive for our manuscript.

Reviewer 2 Report

The topic of the paper is interesting and the paper is very well structured. The novelty of the paper is well stated and the question that authors tried to address in their research has been fully addressed. However, the following are required to be considered by the authors before the paper being accepted. 

·         The current conclusions are massive, and it contains lots of unnecessary information. Therefore, authors are firstly required to state the main aim of the current research at the conclusion before listing the main findings. Then the findings should be summarised in and shortened as they are seems to be too much information there.

Author Response

Dear reviewers:

Thank you very much for your kindly comments on our manuscript. There is no doubt that these comments are valuable and very helpful for revising and improving our manuscript. We have studied comments carefully and have made revision which marked in red in revised paper. We have tried our best to revise our manuscript according to the comments. Attached please find the revised version, which we would like to submit for your kind consideration.

In what follows, we would like to answer the questions you mentioned and give detailed account of the changes made to the original manuscript.

Responses to the reviewer’s comments

Reviewer #2:

  1. Response to comment:

The topic of the paper is interesting and the paper is very well structured. The novelty of the paper is well stated and the question that authors tried to address in their research has been fully addressed. However, the following are required to be considered by the authors before the paper being accepted. 

The current conclusions are massive, and it contains lots of unnecessary information. Therefore, authors are firstly required to state the main aim of the current research at the conclusion before listing the main findings. Then the findings should be summarized in and shortened as they are seemed to be too much information there.

Response:

We thank you for your careful reading of our paper and give this suggestion. According to your suggestion. The conclusion of our manuscript has been summarized and shortened. The details of the modifications are as follows:

In this paper, a FEM-SPH coupling model was developed to analyze the large deformation problem caused by the installation of screw pile. The screw pile was modelled using the finite element method and the soil model was built using the SPH method, the installation process of screw pile is well simulated. And the FEM-SPH simulation results were compared with the field tests results and traditional finite element simulation result. In addition, based on the established FEM-SPH numerical model, the differences of different screw piles were analyzed during the installation process. Based on the analyses conducted in this paper, the following conclusions can be obtained:

(1) By comparing with the experimental results, the accuracy of the FEM-SPH coupling model is verified. FEM-SPH simulation results shows a better agreement with the experimental results in terms of the installation torque and the soil deformation of ground surface, indicated the FEM-SPH method has a good applicability in dealing with the large deformation caused by the installation of screw pile.

(2) Compared to the traditional FE (AlE) method, the FEM-SPH method possesses better efficiency and accuracy at the same time, which is more suitable for simulating the large deformation problem caused by the installation of screw pile.

(3) The FEM-SPH numerical model was used to analyze the installation process of the screw pile. The numerical results show that the interaction during the installation process between screw pile and the surrounding soil is transferable and the load transfer occurs throughout the installation of the screw pile. With the increasing of the installation depth, the extent of damage to the soil caused by screw pile will decrease. The installation of screw pile results in heave damage to the shallow soil and exhibits cylindrical shear damage to the middle and deeper layers.

(4) Based on the developed numerical model, the variability in the installation process of different screw piles was analyzed. The analysis shows that the two failure mechanisms (HFM and CSFM) have different effects on the soil during the installation of screw pile. For HFM and CSFM caused by the installation of screw pile, the influence range on the soil around the pile of HFM is only related to the pile diameter and irrelevant to the spiral pitch. Unlike HFM, the influence range on the soil around the pile of CSFM is affected by the variations of both spiral pitch and pile diameter.

Finally, thanks again for your valuable comments, which are very important and instructive for our manuscript.

Reviewer 3 Report

Please see the comments below for the improvement of the manuscript:

1. fig 1: doamin should be domain. please pay attention to spelling in English

2. in line 141 in the inline equation, there is a strange symbol, it looks Chinese. it needs to be corrected.

3. line 153: "So the function of the particle ? is approximated can be expressed as:". Use either "is approximated as" or "is expressed as" but not both of them

4. line 235, table: Kg should be kg (lower case k) here and all over the text

5. the same table: Kpa should be kPa. these are standard symbols and the upper/lower case cannot change! the same for Mpa-> MPa and similar. check all over the manuscript for such small errors. see table 3 and others. see fig 7 b and others.

6. some properties appear to be missing from table 2, such as H1, H3 and D2. is that on purpose? is there a reason for that?

7. we see in table 2 that L=1m, but in fig 7b, the depth goes up to 280 mm=0.28 m. why is that? why do you examine only the 28% of the depth and not the full depth? similarly fig 8a,b goes to 650 mm depth.

8. i would suggest that figures that have to do with depth be presented in the vertical axis, such as in fig 7b. why not keep this consistent? we see that this changes in fig 8 where depth is presented in the horizontal axis. keep things consistent and in my opinion it makes more sense to present depth in the vertical axis only

9. fig 10. installation is spelled incorrectly in the vertical axis. please check ALL captions and figures for such spelling mistakes which can be common.

10. what is the unit used in fig 11? I see v-m, what is that? or is it von misses? in this case, please report the unit used

11. what are the numbers in fig 13? the ones on the right such as A 1041533? also the same in other similar figures. why do you need these numbers?

12. in fig 13,14 radial displacement is reported with (m) units, while the units reported in fig 15 for the same are (s). why is that? please fix if there is a mistake in the units.

13. mention the units for the displacement in fig 16 and similar. make sure that units are reported in ALL figures. the same for the pressure in fig 17 and similar.

Author Response

Dear reviewers:

Thank you very much for your kindly comments on our manuscript. There is no doubt that these comments are valuable and very helpful for revising and improving our manuscript. We have studied comments carefully and have made revision which marked in red in revised paper. We have tried our best to revise our manuscript according to the comments. Attached please find the revised version, which we would like to submit for your kind consideration.

In what follows, we would like to answer the questions you mentioned and give detailed account of the changes made to the original manuscript.

Responses to the reviewer’s comments

Reviewer #3:

  1. Response to comment:

fig 1: doamin should be domain. please pay attention to spelling in English

Response:

  We thank you for your reminding us this important point. We have made changes and the figure have been updated(Fig.1 in the revised manuscript).

  1. Response to comment:

in line 141 in the inline equation, there is a strange symbol, it looks Chinese. it needs to be corrected.

Response:

  We thank you for your reminding us this important point, we have made changes(line142 in the revised manuscript).

  1. Response to comment:

  line 153: "So the function of the particle ? is approximated can be expressed as:". Use either "is approximated as" or "is expressed as" but not both of them.

Response:

  We gratefully appreciate for your valuable suggestion. The sentence has been modified: "So the function of the particle ? is approximated as:"( line153 in the revised manuscript).

  1. Response to comment:

  line 235, table: Kg should be kg (lower case k) here and all over the text.

Response:

Thank you so much for your careful check. According to your suggestion, We have checked the spelling in the manuscript and made an effort to correct the spelling errors. These modifications have been marked in red in the manuscript, for example, Kg has been modified to kg (Table.1 in the revised manuscript).

  1. Response to comment:

the same table: Kpa should be kPa. these are standard symbols and the upper/lower case cannot change! the same for Mpa-> MPa and similar. check all over the manuscript for such small errors. see table 3 and others. see fig 7 b and others.

Response:

Thank you so much for your careful check. According to your suggestion, We have checked these standard symbols in the manuscript and made an effort to correct those. These modifications have been marked in red in the manuscript (Table.1, Table.2 and Fig.8(b) in the revised manuscript).

  1. Response to comment:

some properties appear to be missing from table 2, such as H1, H3 and D2. is that on purpose? is there a reason for that?

Response:

We thank you for your careful reading of our paper and ask this question. In fact, these parameters can be solved by other parameters. For example,  can be obtained by  and : . According to your suggestions, in order to provide a better understanding of the article, these specific parameters have been added to table 2.

  1. Response to comment:

we see in table 2 that L=1m, but in fig 7b, the depth goes up to 280 mm=0.28 m. why is that? why do you examine only the 28% of the depth and not the full depth? similarly fig 8a,b goes to 650 mm depth.

Response:

We thank you for your careful reading of our paper and ask this question. As for Fig.7 (Fig.8 in the revised manuscript), the purpose of the numerical CPT test is mainly to preliminarily verify the rationality of the soil model established by SPH method. In order to reduce the calculation time, so we only tested the soil characteristics under the depth of 300mm. In fact, the verification of the soil model is carried out by comparing the numerical results with the field test results (Fig. 9 in the revised manuscript). For Fig.9, the test depth is 700mm, which is determined by the structure of screw pile. As shown in Fig.4 in the manuscript, for this kind of micro screw pile, it is mainly driven into the soil at the part containing spiral blades, and the length of this section is exactly 700mm. For the part without spiral blades, it needs to be located above the ground to withstand horizontal or compressive loads.

  1. Response to comment:

  I would suggest that figures that have to do with depth be presented in the vertical axis, such as in fig 7b. why not keep this consistent? we see that this changes in fig 8 where depth is presented in the horizontal axis. keep things consistent and in my opinion it makes more sense to present depth in the vertical axis only.

Response:

  We gratefully appreciate for your valuable suggestion. According to your suggestion, those figures has been modified in the manuscript. The Specific modifications can be viewed in the revised draft (Fig.9 in the revised manuscript).

  1. Response to comment:

  fig 10. installation is spelled incorrectly in the vertical axis. please check ALL captions and figures for such spelling mistakes which can be common.

Response:

  Thank you so much for your careful check. According to your suggestion, We have modified it and the figure has been updated (Fig.11 in the revised manuscript). And other captions and figures also have been checked in the manuscript.

  1. Response to comment:

  what is the unit used in fig 11? I see v-m, what is that? or is it von misses? in this case, please report the unit used.

Response:

  We gratefully appreciate for your valuable suggestion. In our manuscript, the soil stress was described by using von misses stress, whose unit is Pa. According your suggestion, the unit used has been added in the manuscript (Fig.12 in the revised manuscript).

  1. Response to comment:

  what are the numbers in fig 13? the ones on the right such as A 1041533? also the same in other similar figures. why do you need these numbers?

Response:

We thank you for your careful reading of our paper and ask this question. For the soil model established by SPH method, it contains many particles. In order to better determine the measured range and soil particles in the numerical simulation, so these numbers were used to define the measured soil particles.

  1. Response to comment:

in fig 13,14 radial displacement is reported with (m) units, while the units reported in fig 15 for the same are (s). why is that? please fix if there is a mistake in the units.

Response:

We thank you for your reminding us this important point, we have made changes and the figure has been updated (Fig.16 in the revised manuscript).  

  1. Response to comment:

mention the units for the displacement in fig 16 and similar. make sure that units are reported in ALL figures. the same for the pressure in fig 17 and similar.

Response:

We thank you for your reminding us this important point. According to your suggestion, we have made changes and those figures has been updated (Fig.17, Fig.18 and Fig.19 in the revised manuscript). 

Finally, thanks again for your valuable comments, which are very important and instructive for our manuscript.

Round 2

Reviewer 1 Report

Authors have greatly improved the manuscript and answered all of my remarks. Now I recommend acceptance of the paper in the present form.